# Selective expression of variant surface antigens enables *Plasmodium falciparum* to evade immune clearance in vivo

Marvin Chew[1,2,7], Weijian Ye[1,2,7], Radoslaw Igor Omelianczyk[1], Charisse Flerida Pasaje[3], Regina Hoo[1,6], Qingfeng Chen[4], Jacquin C. Niles[3], Jianzhu Chen[2,5,8✉] & Peter Preiser[1,2,8✉]

*Plasmodium falciparum* has developed extensive mechanisms to evade host immune clearance. Currently, most of our understanding is based on in vitro studies of individual parasite variant surface antigens and how this relates to the processes in vivo is not well-understood. Here, we have used a humanized mouse model to identify parasite factors important for in vivo growth. We show that upregulation of the specific PfEMP1, VAR2CSA, provides the parasite with protection from macrophage phagocytosis and clearance in the humanized mice. Furthermore, parasites adapted to thrive in the humanized mice show reduced NK cell-mediated killing through interaction with the immune inhibitory receptor, LILRB1. Taken together, these findings reveal new insights into the molecular and cellular mechanisms that the parasite utilizes to coordinate immune escape in vivo. Identification and targeting of these specific parasite variant surface antigens crucial for immune evasion provides a unique approach for therapy.

[1] School of Biological Sciences, Nanyang Technological University, Singapore, Singapore. [2] Singapore-MIT Alliance for Research and Technology, Antimicrobial Resistance Interdisciplinary Research Group, Singapore, Singapore. [3] Department of Biological Engineering, Massachusetts Institute of Technology, Cambridge, MA, USA. [4] Humanized Mouse Unit, Institute of Molecular and Cell Biology, Agency of Science, Technology and Research, Singapore, Singapore. [5] Koch Institute for Integrative Cancer Research and Department of Biology, Massachusetts Institute of Technology, Cambridge, MA, USA. [6] Present address: Wellcome Sanger Institute, Hinxton, Cambridgeshire CB101SA, UK. [7] These authors contributed equally: Marvin Chew, Weijian Ye. [8] These authors jointly supervised this work: Jianzhu Chen, Peter Preiser. ✉email: jchen@mit.edu; PRPreiser@ntu.edu.sg

Plasmodium falciparum is the most important causative agent of human malaria. Currently, annual malaria infections and mortality are approximately 220 million and 400,000, respectively[1]. During the erythrocytic stage of malaria infection, the parasite invades red blood cells (RBCs) and expresses variant surface antigens (VSAs) on the surface of the infected red blood cell (iRBC). These VSAs such as P. falciparum Erythrocyte Membrane Protein 1 (PfEMP1)[2] and repetitive interspersed family of polypeptides (RIFINs)[3] bind to host receptors and sequester to specific tissues and form rosettes to avoid clearance by the spleen. More recently, these multigenic and highly polymorphic VSAs, have been shown to play a role in evading recognition by the host innate immune system[4].

Macrophages and natural killer (NK) cells are the earliest innate immune cells that respond to parasite infection[5,6], and the outcome of this early host–parasite interaction is a strong determinant for immunopathology and disease severity[7,8]. However, parasites have developed mechanisms to inhibit macrophage phagocytosis[9,10] and evade killing by NK cells[4,8]. Macrophages recognize and phagocytose parasite-infected red blood cells (iRBC) through the class B scavenger receptor, CD36, which binds to group B and C PfEMP1 expressed on the surface of iRBC[11]. Parasites isolated from patients with severe malaria exhibit reduced expression of PfEMP1 that binds to CD36[12]. A subset of RIFIN has also been shown to inhibit NK cell activation through leukocyte immunoglobulin-like receptor subfamily B member 1 (LILRB1)[4]. Despite this progress, the determinants that allow a parasite to evade macrophage and NK cell clearance and thrive in the human host have not been identified.

One of the main challenges of immunological studies of P. falciparum is the lack of a suitable model system that accurately emulates all the intricacies of the complex immune responses the human host develops in response to an infection. Current studies using in vitro cultured parasites are further hampered by the fact that through continuous in vitro culture, numerous parasitic adaptations have occurred, including the loss of parasitic virulence factors[13–16]. Noticeably, there is an overall downregulation of PfEMP1[17] and specific downregulation of the non-CD36-binding Group A var genes[18]. Interestingly, parasites isolated from patients with severe malaria expressed a higher level of group A var transcripts compared to parasites isolated from patients with non-severe malaria[12]. Similarly, expression of RIFIN as well as STEVOR, two additional families of variant surface antigens, are absent or expressed at a lower level in continuous in vitro cultured strains compared to parasites directly isolated from patients[19,20].

To gain a better understanding in the mechanisms the parasite utilizes to effectively survive in vivo, we characterized the changes that occur when in vitro cultured P. falciparum is adapted to grow in human RBC-reconstituted NOD/SCID IL2γ[null] (RBC-NSG) mice. In RBC-NSG mice, human RBCs support parasite infection in the presence of mouse macrophages as NSG mice are deficient in T, B, and natural killer (NK) cells. The system provides a reductionist in vivo environment to study the changes the parasites need to undergo to evade this first line of defense of the innate immune system. We show that in the presence of physical and immune stress, parasites upregulate the expression of specific var and rifs. Specifically, we identify that upregulation of VAR2CSA, a member of PfEMP1 that does not bind to CD36, appears to play a critical role in enabling parasites to escape from macrophage phagocytosis. These adapted parasites also demonstrated enhanced immune evasion of NK cell-mediated killing through the interaction with the immune suppressive receptor, LILRB1. Our findings reveal molecular and cellular mechanisms involved in parasite adaptation to immunological stress.

## Results

**Patent infection of RBC-NSG mice by P. falciparum requires a period of adaptation**. To study P. falciparum adaptation to immunological and physical stresses in vivo, we infected RBC-NSG mice with an equal inoculum of six different P. falciparum strains that had been continuously cultured in vitro (Table 1). In all cases, no parasites were detected by Giemsa staining of peripheral blood smears until at least day 18 after the initial inoculation (Fig. 1a), consistent with a previous report[21]. By 35 days after the initial inoculation when parasitemia was apparent, blood was collected from the infected mice and used to infect human RBCs in vitro. After at least 20 cycles of in vitro culture and expansion, the recovered parasites were used to infect a new batch of RBC-NSG mice. No delay in parasitemia was observed this time around (Fig. 1a). Similarly, when whole blood, including

**Table 1 List of parasites adapted and used.**

| Parasite | Source | Description | Day post infection for adapted parasite to appear in peripheral blood |
|---|---|---|---|
| 3D7 | BEI Resources (MRA-102) | Clone derived from NF54 by limiting dilution[61]. | 3D7-B7 AS – Day 27 |
| W2mef | BEI Resources (MRA-615) | Mefloquine-resistant culture lines derived from Indochina strain[62]. | W2mef AS – Day 19 |
| T994 | BEI Resources (MRA-153) | Isolate derived from T9 strain collected from patient in Thailand[63]. | T994 AS – Day 21 |
| NF54attB | Niles lab | NF54attB that harbors an attB site at the Cg6 locus con Chr 7[64]. | NF54attB AS – Day 24 |
| 3D7attB | BEI Resources (MRA-845) | 3D7 parasite that harbors an attB site at the Cg6 locus on Chr 7[23]. | 3D7attB AS – Day 30 |
| NF54attB[Cas9+T7 Polymerase] (NF54CR) | Niles lab | NF54 parasites that expressed Cas9 and T7 RNA polymerase[65]. | NF54CR AS – Day 25 |
| NF54CR AS mahrp1 | This paper | mahrp1 gene under the regulation of aTc on the NF54CR AS background. | Adapted parasites used for transfection |
| 3D7-SLI-RIFIN* | This paper | 3D7 strain with PF3D7_1254800 gene tagged by a resistant marker for continuous expression. | N.A. |
| 3D7-SLI-PfEMP1* | This paper | 3D7 strain with PF3D7_0421300 gene tagged by a resistant marker for continuous expression. | N.A. |

*N.A. Not adapted

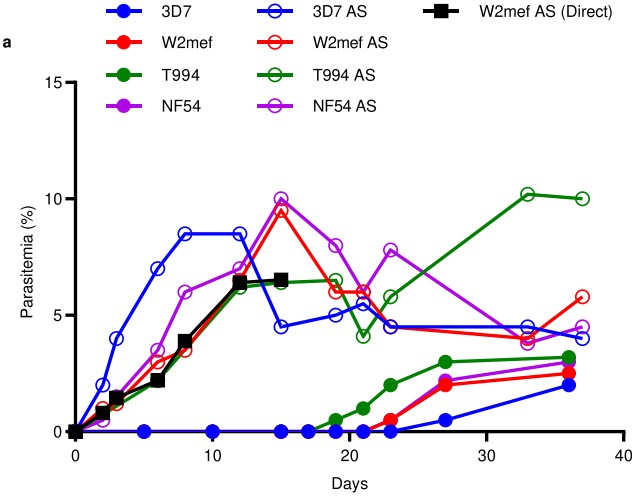

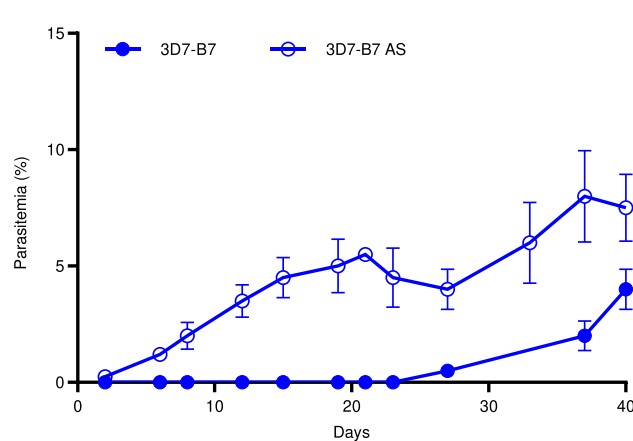

**Fig. 1 Comparison of parasitemia of non-adapted and adapted _P. falciparum_ strains in RBC-NSG mice. a** RBC-NSG mice were infected with the indicated strains of non-adapted or adapted (AS) _P. falciparum_ strains, and parasitemia was assessed by Giemsa staining of peripheral blood smears. Black line shows parasitemia levels (tracked till Day 15) of RBC-NSG mice directly infected with whole blood, including iRBCs, from RBC-NSG mice that had been infected with non-adpated 3D7 parasite 25 day earlier and was parasitemic. Each point represents the parasitemia of each mouse's respective parasite strain. **b** A single clone of 3D7, 3D7-B7, was isolated, expanded, and used to infect RBC-NSG mice. After 40 days, parasites from peripheral blood were recovered, cultured, and expanded in vitro to generate 3D7-B7 AS. 3D7-B7 AS parasites were then used to infect a new batch of RBC-NSG mice. Each point represents the mean ± SD of _n_ = 3 biologically independent experiments. Source data are provided as a Source Data file.

iRBCs, was taken from RBC-NSG mice and used directly to infect a new batch of RBC-NSG mice, no delay in parasite growth was observed (black line in Fig. 1a). These results show that the parasites have undergone adaptation in the RBC-NSG mice and have acquired a phenotype that is competent in infecting the RBC-NSG mice and can be stably maintained in vitro culture for at least 20 cycles.

Since long-term continuous in vitro culture of _P. falciparum_ leads to genetic heterogeneity[22], we serially diluted the non-adapted 3D7 strain of _P. falciparum_ to obtain a single clone, 3D7-B7, and repeated the adaptation experiment in RBC-NSG mice. Once again, there was a significant delay of 26 ± 5.1 days before 3D7-B7 was detected in the peripheral blood circulation of

inoculated mice (Fig. 1b). Similarly, when the recovered parasites from RBC-NSG mice were cultured in vitro for 20 cycles and subsequently used to infect RBC-NSG mice, no delay in parasitemia was observed. Therefore, the observed parasite adaptation in RBC-NSG mice is not due to the selection of a single unique parasite from a genetically heterogeneous culture. Parasites recovered from RBC-NSG mice after adaptation are referred to as adapted strains (AS).

**Adaptation of _P. falciparum_ parasites is associated with transcriptional changes of specific variant surface antigens**. To understand the changes contributing to the adaptation phenotype, we determined the genetic differences between three non-adapted and adapted parasites (3D7-B7, 3D7attB, and NF54attB) by whole genome sequencing[23]. Single nucleotide polymorphism (SNP) variant calling identified similar SNPs in both the parental non-adapted strains and their corresponding adapted strains and no conserved unique SNP was detected among the three adapted strains. Kinship coefficient analysis[24] of the SNPs called for each parasite strain pair did not significantly diverge and had high kinship coefficient values (>0.487, Fig. 2a). Since kinship coefficient values of above 0.354 corresponds to duplicate or monozygotic twins[25], our results suggest that adaptation is not due to the acquisition of a unique mutation(s).

We next examined whether transcriptional changes could be responsible for the adaptation phenotype by comparing the gene expression profile of three pairs of non-adapted and adapted _P. falciparum_ strains (3D7-B7, 3D7attB, and NF54CR) using microarray analysis[26]. We included NF54CR, a strain of NF54attB parasites stably expressing Cas9 and T7 RNA polymerase[27], as it was used for later studies. Cultures of infected RBCs were tightly synchronized and RNA was harvested every 8 h across the intraerythrocytic development cycle (IDC) to generate six time points per parasite strain. Significance Analysis of Microarray (SAM)[28] of the 3 pairs of non-adapted and adapted parasite strains using a delta of 0.55 showed that 27 genes and 93 genes were significantly upregulated and downregulated, respectively (Fig. 2b and Supplementary Fig. 1). Using a false discovery rate (FDR) of 0.01, we further refined these genes to derive a list of differentially expressed genes (DEG), of which 14 were upregulated and 36 were downregulated (Fig. 2c and Tables 2 and 3). Gene Ontology (GO)[29,30] biological processes analysis[31] revealed that the DEGs are involved in processes such as modulation of erythrocyte aggregation, antigenic variation, and cytoadherence to microvasculature (Fig. 2d). Many of the DEGs were parasite variant surface antigens, including one _var_ gene and 3 _rif_ genes that were upregulated and 4 _var_ genes, 5 _rif_ genes, 4 _stevor_ genes and 1 _surf_ gene that were downregulated (Fig. 2c and Tables 2 and 3).

To understand the transcriptional differences of VSAs, we further analyzed the time-course expression of the differentially expressed _var_, _rif_ and _stevor_. In non-adapted parasites, the _var_ genes with highest expression differed from each other (PF3D7_1240900 in 3D7-B7, PF3D7_0412400 in 3D7attB, and PF3D7_0420700 in NF54CR) (Table 4), consistent with a previous report[17]. These three _var_ genes belongs to groups B or C var genes and are known to interact with CD36. In contrast, in all three adapted strains, the same _var_ gene PF3D7_1200600 (_var2csa_) was the most highly expressed (Table 4 and Supplementary Fig. 1a). _Var2csa_ belongs to group E PfEMP1 and does not interact with CD36[32]. Furthermore, 4 _var_ genes that belong to group C[33] were significantly downregulated in adapted parasites (Supplementary Fig. 1). Overall, the expression of these differentially expressed _var_ genes followed a temporal profile where the transcripts level peaked at 16 h post infection (hpi), and

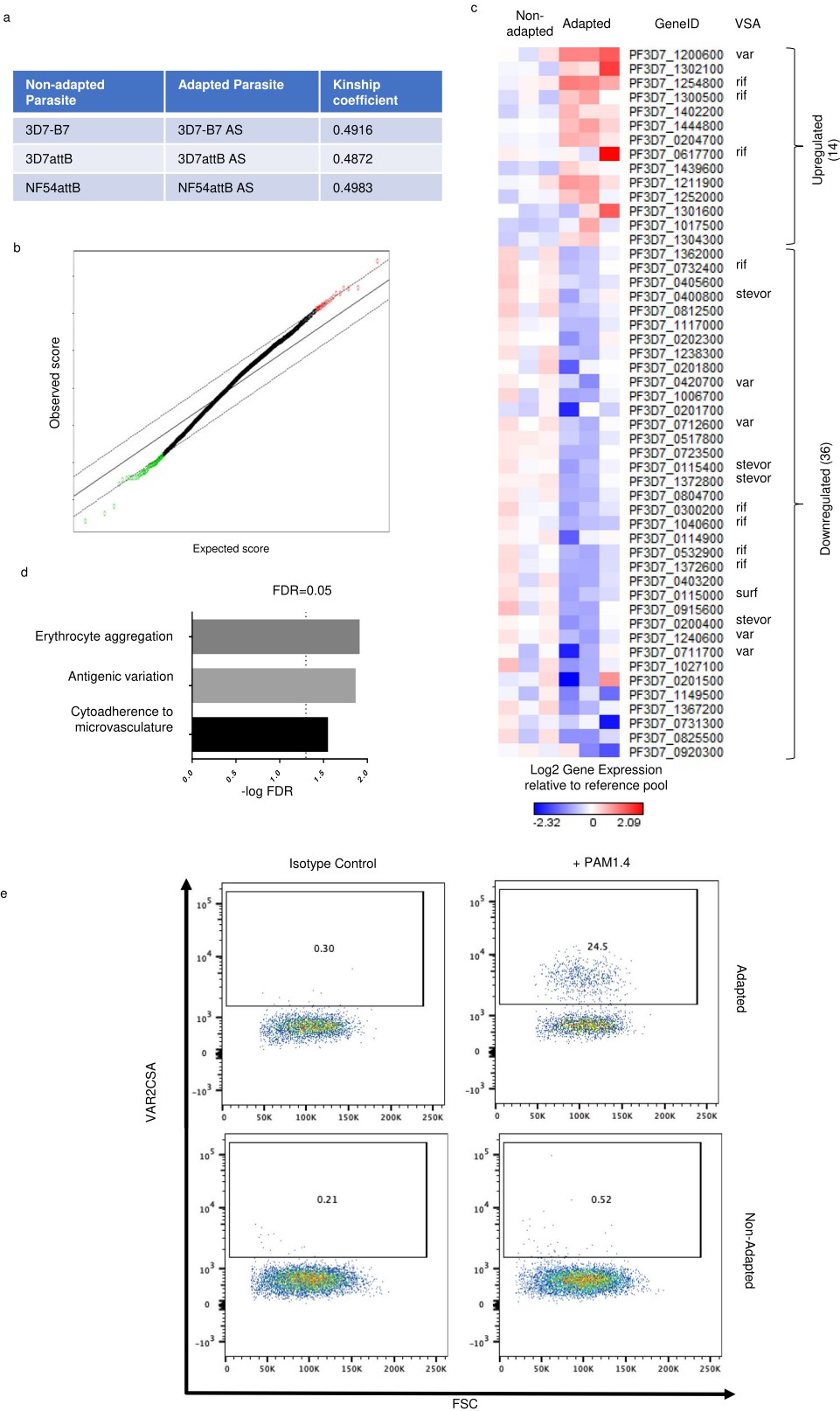

**Fig. 2 Differences in gene expression between adapted and non-adapted parasites. a** Kinship coefficient analysis of SNPs in adapted and non-adapted parasites. **b** SAM of the 3 pairs of non-adapted and adapted parasite strains using a delta of 0.55. Red: upregulated genes; green: downregulated genes. **c** Heatmap of upregulated and downregulated DEGs identified using an FDR of 0.01. Each column represents a different parasite strain. Each row represents a DEG (average over the six different time points). VSAs are identified. Source data are provided as a Source Data file. **d** Top three GO biological processes that were significantly enriched in DEGs. **e** Adapted parasites (top row) and non-adapted parasites (bottom row) were either stained with isotype control antibody or PAM1.4 antibody, followed by FITC-labeled secondary antibody and flow cytometry. Shown are staining profiles of forward scatter (FSC) versus PAM1.4. The numbers indicate percentages of parasites that are positive for PAM1.4 (VAR2CSA).

**Table 2 Upregulated DEGs.**

| Gene ID | Description | Bonferroni p val | log$_2$FC |
|---|---|---|---|
| PF3D7_1200600 | erythrocyte membrane protein 1, PfEMP1 | 5E-16 | 1.139171 |
| PF3D7_1302100 | gamete antigen 27/25 | 3.4E-11 | 0.93061 |
| PF3D7_1254800 | rifin | 2.92E-10 | 0.884545 |
| PF3D7_1300500 | rifin | 2.77E-06 | 0.657873 |
| PF3D7_1402200 | cytochrome c oxidase subunit ApiCOX19, putative | 2.05E-05 | 0.597823 |
| PF3D7_1444800 | fructose-bisphosphate aldolase | 2.37E-05 | 0.593211 |
| PF3D7_0204700 | hexose transporter | 2.72E-05 | 0.5889 |
| PF3D7_0617700 | rifin | 3.14E-05 | 0.584273 |
| PF3D7_1439600 | cytochrome c oxidase subunit ApiCOX26, putative | 3.29E-05 | 0.582841 |
| PF3D7_1211900 | non-SERCA-type Ca2 + -transporting P-ATPase | 4.41E-05 | 0.573354 |
| PF3D7_1252000 | tRNA Glutamine | 4.77E-05 | 0.57074 |
| PF3D7_1301600 | erythrocyte binding antigen-140 | 4.86E-05 | 0.57017 |
| PF3D7_1017500 | myosin essential light chain ELC | 7.02E-05 | 0.557975 |
| PF3D7_1304300 | conserved Plasmodium protein, unknown function | 9.28E-05 | 0.548597 |

**Table 3 Downregulated DEGs.**

| Gene ID | Description | Bonferroni p val | log$_2$FC |
|---|---|---|---|
| PF3D7_0920300 | conserved Plasmodium protein, unknown function | 1.29E-08 | −0.79847 |
| PF3D7_0825500 | protein KRI1, putative | 1.48E-08 | −0.79512 |
| PF3D7_0731300 | Plasmodium exported protein (PHISTb), unknown function | 2.12E-08 | −0.7864 |
| PF3D7_1367200 | CLAMP domain-containing protein, putative | 3.45E-08 | −0.77445 |
| PF3D7_1149500 | ring-infected erythrocyte surface antigen 2, pseudogene | 1.78E-07 | −0.7329 |
| PF3D7_0201500 | Plasmodium exported protein (hyp9), unknown function | 4.02E-07 | −0.71138 |
| PF3D7_1027100 | U3 small nucleolar ribonucleoprotein protein MPP10, putative | 5.42E-07 | −0.70335 |
| PF3D7_0711700 | erythrocyte membrane protein 1, PfEMP1 | 6.03E-07 | −0.70048 |
| PF3D7_1240600 | erythrocyte membrane protein 1, PfEMP1 | 9.96E-07 | −0.68674 |
| PF3D7_0200400 | stevor | 1.75E-06 | −0.67099 |
| PF3D7_0915600 | conserved Plasmodium protein, unknown function | 3.62E-06 | −0.65018 |
| PF3D7_0115000 | surface-associated interspersed protein 1.3 (SURFIN 1.3) | 4.33E-06 | −0.64498 |
| PF3D7_0403200 | pre-mRNA splicing factor, putative | 4.83E-06 | −0.64177 |
| PF3D7_1372600 | rifin | 5.69E-06 | −0.6369 |
| PF3D7_0532900 | rifin | 7.11E-06 | −0.63029 |
| PF3D7_0114900 | Plasmodium exported protein, unknown function, pseudogene | 1.48E-05 | −0.60804 |
| PF3D7_1040600 | rifin | 1.51E-05 | −0.60739 |
| PF3D7_0300200 | rifin | 1.64E-05 | −0.60488 |
| PF3D7_0804700 | conserved Plasmodium protein, unknown function | 1.68E-05 | −0.6041 |
| PF3D7_1372800 | stevor | 2.06E-05 | −0.59764 |
| PF3D7_0115400 | stevor | 2.16E-05 | −0.59622 |
| PF3D7_0723500 | dynactin subunit 5, putative | 2.38E-05 | −0.59313 |
| PF3D7_0517800 | apicortin, putative | 3.42E-05 | −0.58156 |
| PF3D7_0712600 | erythrocyte membrane protein 1, PfEMP1 | 3.55E-05 | −0.58037 |
| PF3D7_0201700 | DnaJ protein, putative | 4.09E-05 | −0.57578 |
| PF3D7_1006700 | conserved Plasmodium protein, unknown function | 4.29E-05 | −0.57426 |
| PF3D7_0420700 | erythrocyte membrane protein 1, PfEMP1 | 4.36E-05 | −0.5737 |
| PF3D7_0201800 | knob associated heat shock protein 40 | 4.44E-05 | −0.57314 |
| PF3D7_1238300 | pre-mRNA-splicing factor CWC22, putative | 5.95E-05 | −0.56352 |
| PF3D7_0202300 | Plasmodium exported protein (hyp11), unknown function | 6.44E-05 | −0.56085 |
| PF3D7_1117000 | conserved Plasmodium membrane protein, unknown function | 6.61E-05 | −0.55999 |
| PF3D7_0812500 | RNA-binding protein, putative | 6.74E-05 | −0.55936 |
| PF3D7_0400800 | stevor | 7.76E-05 | −0.55462 |
| PF3D7_0405600 | TMEM33 domain-containing protein, putative | 8.7E-05 | −0.55078 |
| PF3D7_0732400 | rifin | 8.88E-05 | −0.55009 |
| PF3D7_1362000 | cytochrome c oxidase subunit ApiCOX24, putative | 9.32E-05 | −0.54845 |

then dipped to a minimum at 40 hpi (Fig. 3a–c, Supplementary Fig. 1b–f), in agreement with previous observations[34].

We performed qRT-PCR analysis of the *var* gene expression[35] of in vitro cultured non-adapted 3D7 parasites, adapted 3D7 parasites that were obtained directly from infected RBC-NSG mice, and the adapted parasites that were cultured in vitro for 2 weeks (Supplementary Table 1). Results from the qPCR analysis recapitulates the microarray analysis that non-adapted parasites

primarily expressed *var* group B and C genes. Adapted parasites that were freshly isolated from RBC-NSG mice showed high *var2csa* expression (Supplementary Fig. 1g). After culturing for 2 weeks in vitro, non-adapted parasites still showed *var* group B and C gene expression with negligible *var2csa* expression whereas adapted parasites maintained high *var2csa* expression. We also stained non-adapted and adapted 3D7 parasites with the well-characterized anti-VAR2CSA antibody, PAM1.4[36] followed by

flow cytometry (Supplementary Fig. 7). As shown in Fig. 2e, about 25% of the adapted parasites stained positive whereas only background level of non-adapted parasites was positive.

**Table 4 Most upregulated var in non-adapted and adapted parasites.**

| Adaptation | Strain | Gene ID | Group | log₂FC |
|---|---|---|---|---|
| Non-adapted | 3D7-B7 | PF3D7_1240900 | C | 1.16 |
| | 3D7attB | PF3D7_0412400 | B | 0.80 |
| | NF54 | PF3D7_0420700 | C | 1.73 |
| Adapted | 3D7-B7 AS | PF3D7_1200600 | var2 | 2.33 |
| | 3D7attB AS | PF3D7_1200600 | var2 | 1.95 |
| | NF54 AS | PF3D7_1200600 | var2 | 2.33 |

Among the *rif* VSA family, 3 type A *rif* were significantly upregulated while 2 were significantly downregulated (Supplementary Fig. 2a). Three type B *rif* were also significantly downregulated (Supplementary Fig. 2b)[37]. Unlike the *var* genes which exhibited a defined maximum and minimum peak of expression, no such temporal pattern was observed for *rif* genes (Fig. 3d, e and Supplementary Fig. 2c–j), consistent with previous observations[38]. In addition, 4 *stevor* genes were significantly downregulated in adapted parasites (Supplementary Fig. 3a–e). Like the *rif* genes, the *stevor* genes were continually expressed throughout the IDC with no visible maximal and minimal peak. Taken together, these results show that adaptation is associated with changes in a relatively small number of genes, many of which are the parasite variant surface antigens (VSAs).

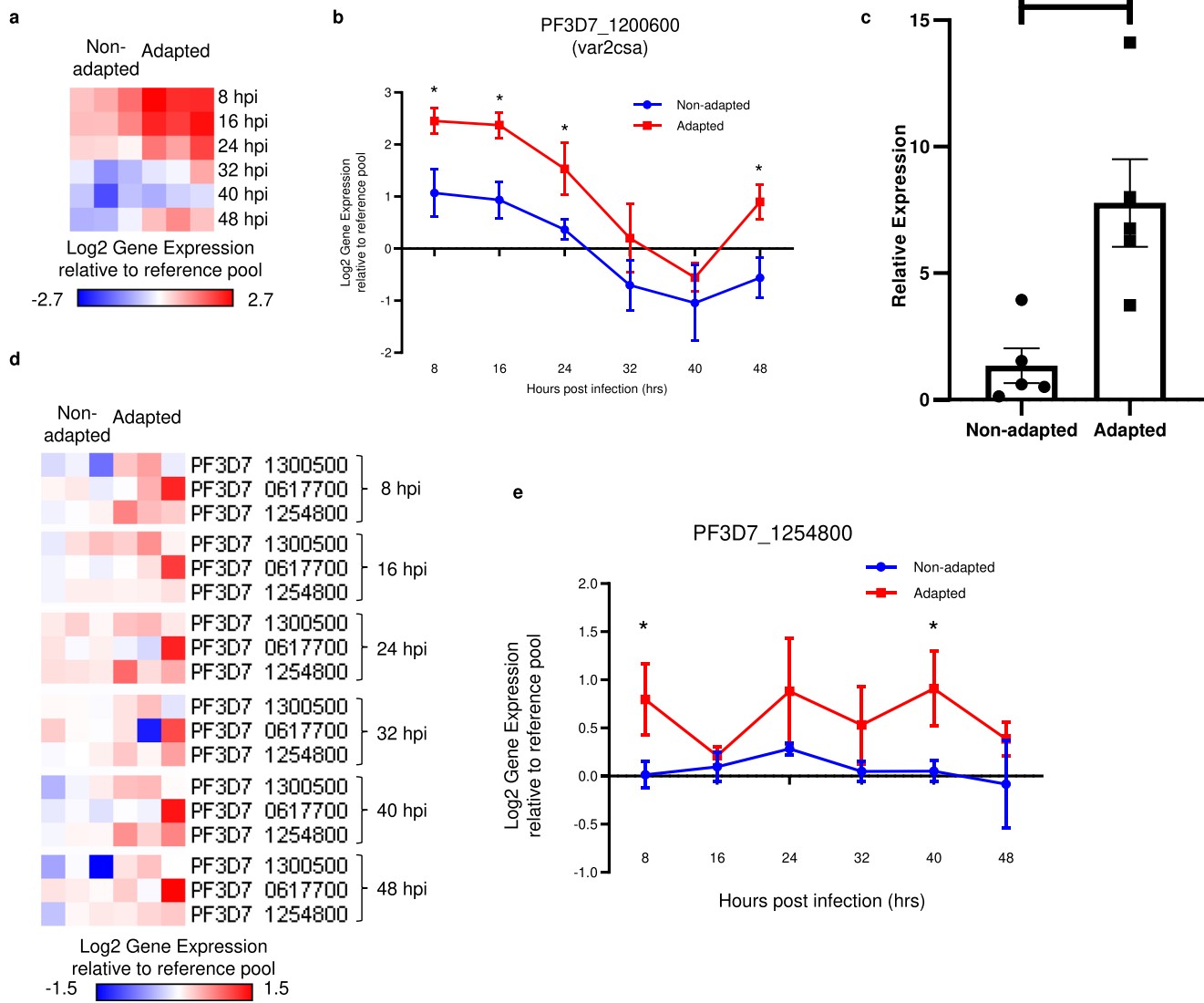

**Fig. 3 Differentially expressed *var* and *rif* genes between adapted and non-adapted parasites. a** Heatmap of upregulated *var* gene (PF3D7_1200600, *var2csa*). Each column represents a different strain of parasite, while each row represents a different time point of the IDC. **b** Temporal gene expression profiles of PF3D7_1200600 across the IDC. Each point represents the mean ± SD. *p < 0.05; Holm-Sidak method for multiple *t* tests (two-tailed). P values are 0.00264 at 8hpi, 0.00234 at 16hpi, 0.00929 at 24hpi, 0.00234 at 48hpi. **c** qPCR validation of average PF3D7_1200600 expression at 16 h post infection where the median expression is 6.77 ± 3.87 (range of 3.73–14.12) for adapted parasites compared to 1.34 ± 1.54 (range of 0.13–3.94) for non-adapted parasites, n = 5 biologically independent samples, mean ± SD. **p = 0.0086, two-tailed *t* test. **d** Heatmap of upregulated *rif* genes. Each column represents a different strain of parasite, while each row represents a different *rif* gene. Each cluster represents a different time point of the IDC. **e** Temporal gene expression profiles of PF3D7_1254800 across the IDC. Each point represents the mean ± SD. *p < 0.05; Holm-Sidak method for multiple *t* tests (two-tailed). P values are 0.018 at 8 hpi, 0.01 at 40 hpi. Source data are provided as a Source Data file.

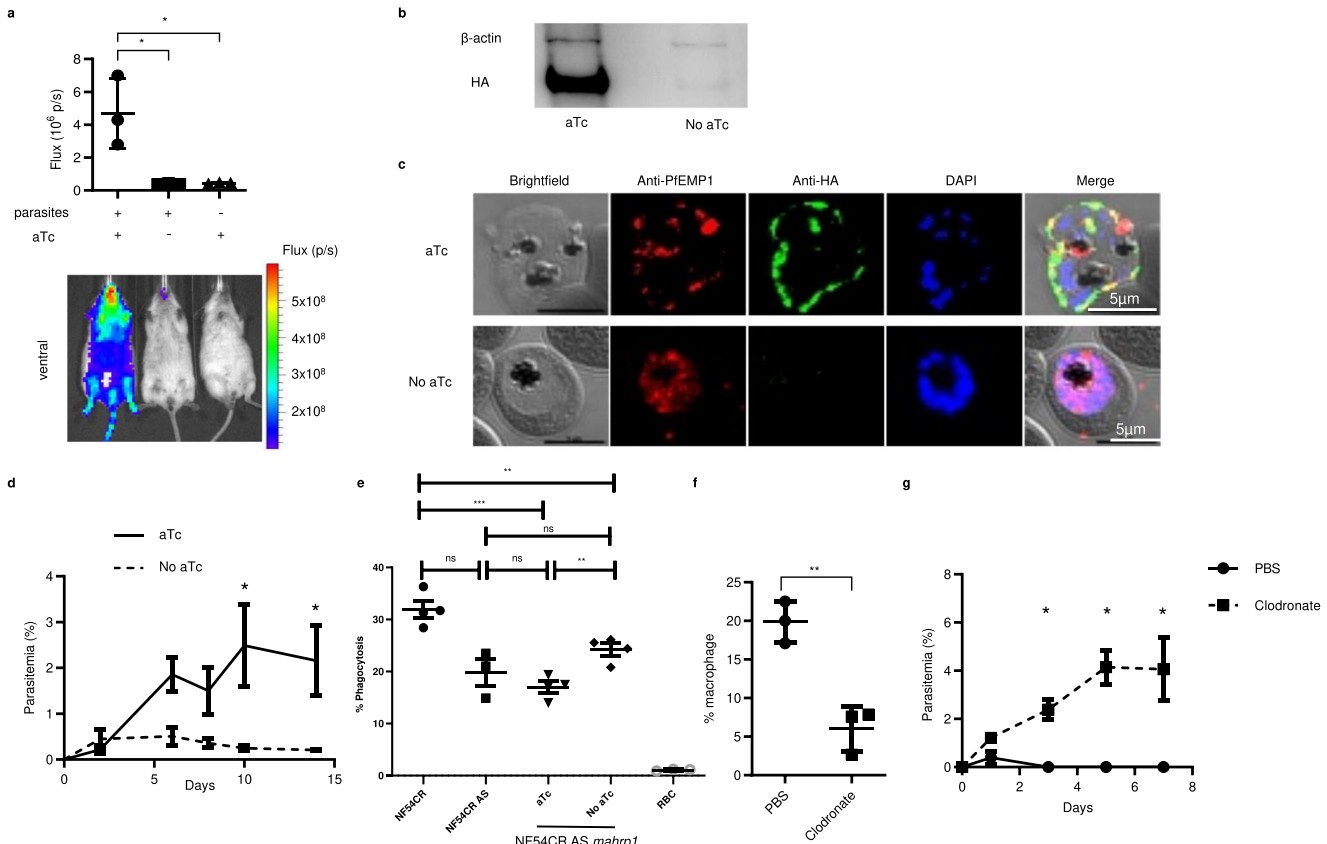

**Fig. 4 Surface expression of VAR2CSA is required for in vivo adaptation. a** RBC-NSG mice were infected with 3D7attB AS pMG56 parasites and were either given aTc or PBS. When parasitemia reached 5%, in vivo IVIS imaging to detect luminescence was performed. Data shown are mean ± SEM, $n = 3$ biologically independent animals. $^*p < 0.03$, One-way ANOVA with Tukey's multiple comparisons. **b** Western blot of HA in NF54CR AS *mahrp1* with or without aTc treatment. Two independent repeats gave similar results. **c** Immunofluorescent assay of NF54CR AS *mahrp1* parasites cultured with or without aTc. Parasites were probed with anti-PfEMP1 (red), anti-HA (green) and DAPI (blue). Scale bar = 5 µm. Two independent repeats gave similar results. **d** Parasitemia of RBC-NSG mice infected with NF54CR AS *mahrp1* with (solid) or without (dash) aTc treatment. Parasitemia was assessed by peripheral blood smear with Giemsa staining. Shown are mean parasitemia ± SEM, $n = 3$ biologically independent animals. $^*p = 0.003$ (Day 10), $^*p = 0.009$ (Day 14), Multiple $T$ test with Holm-Sidak correction. **e** Comparison of macrophage phagocytosis of NF54CR, NF54CR AS, NF54CR AS *mahrp1* with and without aTc treatment of parasites. Data shown are mean ± SEM, $n = 4$ biologically independent samples ($n = 3$ biologically independent samples for NF54CR AS). $^*p < 0.02$, $^{***}p = 0.007$, $^{****}p < 0.0001$. One-way ANOVA with Tukey's multiple comparisons. **f** Percentages F4/80$^{hi}$ CD11b$^{int}$ macrophages quantified by flow cytometry in the spleen of RBC-NSG mice given PBS or chlodronate liposone. Data shown are mean ± SEM, $n = 3$ biologically independent animals, two-tailed $t$ test, $^{**}p = 0.0039$. **g** Parasitemia of RBC-NSG mice infected with 3D7attB in clodronate-treated mice and control mice treated with PBS-liposome. Parasitemia was assessed by peripheral blood smear with Giemsa staining. Data shown are mean ± SEM, $n = 3$ biologically independent animals. $^*p = 0.0029$ (Day 3), $^*p < 0.0001$ (Day 5), $^*p < 0.0001$ (Day 7). Source data are provided as a Source Data file.

**Surface expression of VAS2CSA is required for in vivo adaptation through macrophage evasion.** To determine the roles of identified DEGs in adaptation of parasites in RBC-NSG mice, we adapted the in vitro conditional knockdown TetR-DOZI aptamer system for in vivo regulation of *P. falciparum* genes[39]. To validate this approach, 3D7attB AS parasites were transfected with the pMG56 plasmid encoding the firefly luciferase (FLuc) under the translational control of the TetR-DOZI RNA aptamer system, as well as attP sites for Bxb1 integrase-mediated recombination (Supplementary Fig. 4a), to generate 3D7attB AS pMG56 parasites. Human RBCs were infected with 3D7attB AS pMG56 parasites, serially diluted, and incubated with luciferin with or without aTc. A luminescent signal was readily detected when $5×10^6$ iRBCs were used (Supplementary Fig. 4b). 3D7attB AS pMG56 parasites were used to infect RBC-NSG mice. When parasitemia in infected mice reached 5%, in vivo bioluminescent imaging was performed. Approximately ten-fold higher luminescent signal was detected in aTc treated mice ($4.7 ± 2.1 ×10^6$ p/s) than in mice without aTc treatment ($0.41 ± 0.03 ×10^6$ p/s) or uninfected RBC-NSG given aTc ($0.44 ± 0.04 × 10^6$ p/s (Fig. 4a). These results show that the TetR-

DOZI RNA aptamer system can be utilized to regulate the expression of *P. falciparum* proteins in RBC-NSG mice.

To investigate further the biological implication of *var2csa* upregulation in parasite adaptation in RBC-NSG mice, we attempted to target the TetR-DOZI RNA aptamer system into the *var2csa* locus. This approach failed probably because of the high homology among *var* genes at the 3' ATS segment. Therefore, we used an alternative approach by taking advantage of the fact that members of the PfEMP1 undergo allelic exclusive expression[40]. We tagged membrane-associated histidine rich protein 1 (MAHRP1, PF3D7_1370300), which is required for PfEMP1 surface expression but not required for export of STEVOR and other parasite proteins to the erythrocyte surface[41,42], with the TetR-DOZI system to obtain NF54CR AS *mahrp1* conditional knockdown parasites (Supplementary Fig. 4c). Expression of MAHRP1 was reduced by about 95% in NF54CR AS *mahrp1* in the absence of aTc as compared to the presence of aTc (Fig. 4b). Immunofluorescence analyses revealed punctate staining pattern of PfEMP1 close to the surface of the erythrocyte membrane in the presence of aTc (Fig. 4c) but a more

diffuse staining in the absence of aTc, consistent with an accumulation of PfEMP1 within the parasite plasma membrane, in agreement with previous report[41]. No growth defects in these parasites were observed in continuous culture. Mice were infected with NF54CR AS *mahrp1* parasites that had been cultured in the presence of aTc. An initial parasitemia was observed in all infected mice on day 2, which continued to increase in mice that were given aTC (Fig. 4d). In contrast, parasitemia in mice without aTc treatment did not increase and eventually reduced to background level. These results suggest that adapted parasites require surface expression of VAR2CSA in order to survive and multiply in RBC-NSG mice.

To investigate the cellular mechanisms involved in VAR2CSA-mediated adaptation, we co-cultured NF54CR, NF54CR AS, and NF54CR AS *mahrp1* parasites with human monocyte-derived macrophages (MDM) (Supplementary Fig. 8). A phagocytosis rate of $31.9 \pm 1.6\%$ was observed for non-adapted NF54CR parasites as compared to $19.8 \pm 2.5\%$ with NF54CR AS parasites (Fig. 4e). In the presence of aTc, the phagocytosis rate of NF54CR AS *mahrp1* was $17.0 \pm 1.1\%$, which was not significantly different from NF54CR AS parasites. However, in the absence of aTc, the phagocytosis rate was significantly increased to $24.2 \pm 1.2\%$ (Fig. 4e). Similarly, when murine macrophage cell line RAW 264.7 was used for phagocytosis assay, phagocytosis of the adapted parasites was significantly less than the non-adapted parasites (Supplementary Fig. 5). Consistently, anti-CD36 antibody blocked phagocytosis of non-adapted parasites but not the adapted parasites by human monocyte-derived macrophages (Supplementary Fig. 6). These results suggest that adapted parasites can evade macrophage phagocytosis, probably in part due to selective expression of VAR2CSA that does not bind to CD36[10].

To further verify the involvement of macrophages in the adaptation of *P. falciparum* parasites, we depleted macrophages in RBC-NSG mice with clodronate liposome[43]. After 2 rounds of treatment, the percentage of F4/80hi CD11bint macrophages in the splenic CD45+ population was reduced from $19.9 \pm 2.7\%$ to $6.1 \pm 2.9\%$ (Fig. 4f and Supplementary Fig. 9). The chlodronate-treated mice were infected with non-adapted 3D7 parasites and given chlodronate liposome every 2 days for the duration of the experiment. In clodronate-treated mice, parasitemia rose to $4.1 \pm 2.3\%$ by day 5 while in the untreated mice, no parasites were detectable after day 1 (Fig. 4g). Thus, macrophages are a critical immune cell type that parasites must evade during adaptation in RBC-NSG mice.

**Upregulation of immune-modulating RIFINs in adapted parasites leads to decreased NK cell killing but does not affect macrophage phagocytosis.** We and others have shown that human NK cells can respond to infected RBC directly, although the response is heterogenous among the human population[6,8]. To determine if adapted parasites can resist NK cell-mediated elimination, we screened for and used human donor NK cells that can respond to iRBC. Non-adapted 3D7 or adapted 3D7 AS parasites were co-cultured with NK cells purified from human peripheral blood (Supplementary Fig. 10). While NK cells were able to reduce 3D7 parasitemia by $67.5 \pm 3.2\%$, this reduction was significantly lowered to $44.3 \pm 2.5\%$ for 3D7 AS (Fig. 5a), demonstrating that adapted parasites were able to evade NK cell killing in vitro.

One of the significantly upregulated *rif* gene in our microarray analysis was PF3D7_1254800 (Fig. 3e, f), which had been shown to bind LILRB1-Fc fusion protein and inhibit NK cell activation[4]. To determine if parasites evade NK cells through LILRB1, we repeated the co-culture of NK cells with either 3D7 or 3D7 AS parasites in the presence of an anti-LILRB1 neutralizing antibody

or an isotype control antibody. As shown in Fig. 5b, when NK cells were treated with anti-LILRB1 neutralizing antibody, the parasitemia reduction difference between 3D7 and 3D7 AS was abrogated ($63.9 \pm 4.7\%$ vs $54.5\% \pm 3.8$). Similarly, when compared to isotype control, the presence of the anti-LILRB1 neutralizing antibody enhanced 3D7 AS parasite elimination by NK cells from $39.7 \pm 1.7\%$ to $54.5 \pm 3.8\%$.

We have utilized a selection-linked integration (SLI) approach[44] to select for non-adapted 3D7 parasites that expressed PF3D7_1254800 (3D7-SLI-RIFIN) and verified its role in modulating NK cell-mediated killing[45]. To evaluate whether PF3D7_1254800 also plays a role in reducing phagocytosis by macrophages we utilized the 3D7-SLI-RIFIN along with a parasite line that expressed the SLI tagged *var* gene (PF3D7_0421300) to produce 3D7-SLI-PfEMP1 parasites. Macrophage phagocytosis of the non-adapted 3D7 parasites ($30.3 \pm 2.6\%$), 3D7-SLI-RIFIN ($27.6 \pm 3.5\%$), and 3D7-SLI-PfEMP1 ($25.9 \pm 0.7\%$) was similar, whereas phagocytosis of 3D7 AS ($17.3 \pm 1.9\%$) was significantly reduced (Fig. 5c). These results show that the adapted parasites evade NK cell killing through interaction with LILRB1. However, upregulation of RIFIN PF3D7_1254800 does not reduce macrophage phagocytosis of non-adapted parasites.

## Discussion

Over the course of co-evolution with its human host, *P. falciparum* has developed a repertoire of mechanisms to evade the host immune clearance. One such mechanism is the switching of its surface-exposed VSA, especially VSAs belonging to the *var* family[46]. Expression of *var* in freshly isolated parasites from patients was observed to be highly coordinated, with a single dominant var gene being expressed at any time[47]. However, such coordination is lost during continuous in vitro culture, resulting in a random non-coordinated expression of *var* genes[48], possibly due to the lack of selective pressure by the host immune system. The overall expression of *var* genes are also downregulated[17]. Here, we report that in vivo adaptation of parasites resulted in unique phenotypic and transcriptomic changes in the malaria parasite. Notably, there was an almost four-fold upregulation of the *var2csa* in adapted parasites, coupled with the downregulation of four CD36-binding Group C *var* genes[18].

*Var2csa* is one of three *vars* that are conserved across the different *P. falciparum* strains[49]. VAR2CSA is the only member of the group E PfEMP1[50] and binds chondroitin sulfate A (CSA) but not CD36[50] and has been implicated in pregnancy-associated malaria due to this characteristic[51]. However, recent controlled human malaria infection trials have shown that VAR2CSA is expressed in non-pregnant volunteers as well[52]. This suggests that VAR2CSA expression could have another function apart from binding to CSA found in the placenta[53]. Studies have also shown that parasites isolated from patients with severe malaria express high levels of group A PfEMP1 that do not bind CD36[12,35], suggesting the role of avoiding recognition by CD36 is a mechanism of parasite evasion of macrophage phagocytosis. We have now demonstrated this mechanism of action beyond the previous correlation. Parasites that successfully adapted to grow in RBC-NSG mice expressed VAR2CSA and VAR2CSA-expressing adapted parasites were resistant to macrophage phagocytosis in vitro. When surface expression of VAR2CSA was inhibited, adapted parasites were efficiently phagocytosed by macrophages in vitro and did not induce patent infection in RBC-NSG mice. We speculate that in the absence of PfEMP1 surface expression, the decreased expression of CD47 in late-stage iRBC compared to uninfected or ring-stage parasites makes them susceptible to phagocytosis as shown in previous work[54]. Downregulation of VAR2CSA on the iRBC in the inducible *mharp1*

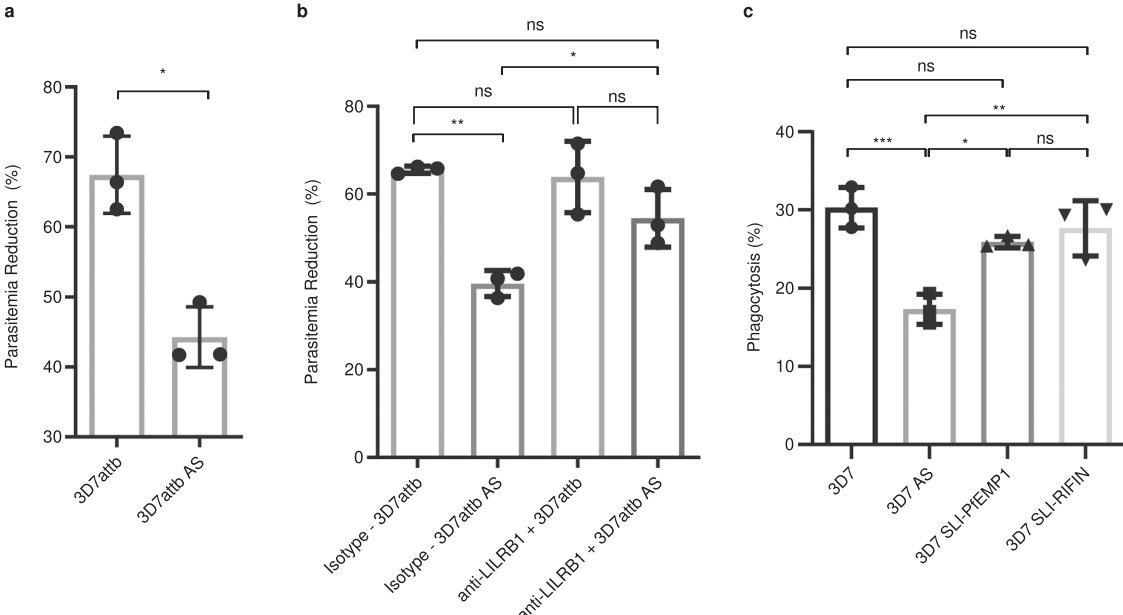

**Fig. 5 LILRB1 inhibits NK cell killing of iRBC but not macrophage phagocytosis. a** NK cells were co-cultured with either non-adapted 3D7attB or adapted 3D7attB AS parasites for 96 hrs and parasitemia was quantified by flow cytometry. Parasitemia reduction was calculated as described in materials and methods. $n = 3$ biologically independent samples, paired $T$ test, *$p = 0.0333$. **b** NK cells were incubated with either an anti-LILRB1 neutralizing antibody or an isotype control and then co-cultured with either 3D7attB or 3D7attB AS parasites for 96 hrs. Parasitemia was quantified by flow cytometry. Data shown are mean ± SD, $n = 3$ biologically independent samples. One-way ANOVA with Tukey's multiple comparisons, *$p = 0.0342$, **$p = 0.0023$. **c** Macrophage phagocytosis assay of 3D7, 3D7 AS, PfEMP1 PF3D7_0421300-expressing 3D7 (3D7-SLI-PfEMP1) parasites and RIFIN PF3D7_1254800-expressing 3D7 (3D7-SLI-RIFIN). Data shown are mean ± SEM, $n = 3$ biologically independent samples, One-way ANOVA with Tukey's multiple comparisons, *$p = 0.0109$, **$p = 0.0036$, ***$p = 0.0008$. Source data are provided as a Source Data file.

knockdown would then expose the infected erythrocyte to CD47-mediated phagocytosis. Conversely, when macrophages were depleted, non-adapted parasites could immediately flourish in RBC-NSG mice. Together, these results demonstrate that the expression of PfEMP1, such as VAR2CSA, that does not bind to CD36, is a mechanism by which parasites evade macrophages in order to establish robust infection in vivo.

RIFINs also showed differential expression during the adaptation process. The type A RIFINs are localized to the iRBC membrane and might be surface-exposed, while type B RIFINs remain inside the infected RBC[37,55]. We found that all upregulated *rif* genes are type A, while three out of five downregulated *rif* are type B. Differential expression between type A and type B RIFINs were also seen in a controlled human malaria infection trial of 4 volunteers where expression of type A RIFINs were upregulated in all volunteers whereas type B RIFINs showed no general changes[56] when compared to in vitro cultured parasites. This suggest that in the RBC-NSG mice and human infection, type A RIFINs are upregulated and expressed as compared to in vitro cultured parasites. Recently, type A RIFINs were shown to bind to inhibitory immune receptors such as LILRB1, LILRB2 and LAIR1 to evade NK and B cell recognition[4,57,58]. Here, we demonstrate that parasites adapted in the RBC-NSG mice show reduced NK cell killing in vitro and upon blocking of LILRB1 receptor with a neutralizing antibody, NK cell killing of the adapted parasite is restored, suggesting the possibility that evasion of NK cell killing by RBC-NSG mice adapted parasites involves RIFINs that are recognized by LILRB1. Notably, our RBC-NSG mice do not have NK, T or B cells, raising the question what is selecting the expression of RIFINs by adapted parasites. In our study, we show that adapted parasites are resistant to elimination by NK cells in vitro and this resistance is abolished when LILRB1 is blocked by antibody. We also showed that elevated expression of PF3D7_1254800 or PF3D7_0421300 does not inhibit

macrophage phagocytosis of non-adapted parasites in vitro. Our results leaves open the possibility that in vivo selection of adapted parasites with increased RIFIN expression in the RBC-NSG mice could still be mediated by macrophages as LILRB1 is predominantly expressed by monocytic lineage cells, NK cells and B cells. Additional studies are required to resolve this issue.

In conclusion, we show that significant changes in transcription occur during parasite adaptation in RBC-NSG mice. These changes are not random but specifically, happen to genes involved in immune evasion. In particular, a single *var* known to be involved in repressing macrophage phagocytosis and a small set of *rif* important to escape NK cell killing was repeatedly upregulated in multiple adapted parasites, suggesting immune evasion during adaptation. As an escape from clearance by the innate immune system is particularly important early on during the establishment of the blood-stage infection when parasite numbers are still relatively small, identification of specific *var* and *rif* that are important during parasite adaptation provide insights into the mechanism of host immune evasion and may lead to improved intervention strategies.

## Methods

***P. falciparum* strains and culturing**. *P. falciparum* blood-stage parasites were cultured in 2.5% hematocrit human RBC in malaria culture media (MCM) comprising of 10.43 g RPMI 1640 powder (Gibco), 25 ml 1 M HEPES (Gibco), 2 g NaHCO$_3$ (Sigma-Aldrich), 5 g Albumax (Gibco), 0.05 g hypoxanthine (Sigma-Aldrich) and 25 mg gentamicin (Gibco) in 1 L milli-Q water. Cultures were incubated in Heracell 150 incubator (Thermo Scientific) at 37 °C in 5% CO$_2$, 3% O$_2$, and 92% N$_2$.

NF54CR (NF54$^{Cas9+T7 Polymerase}$)[27] parasites were kindly shared with us by Prof. Jacquin Niles. Other parasite lines used were obtained from BEI Resources (Table 1).

**Primary cells**. Whole blood was donated by healthy non-malarial immune adult volunteers at the National University Hospital of Singapore Blood Donation Center. Informed consent was obtained from all donors in accordance with approved

protocol and guidelines. Project ethics and approval were obtained from the Institutional Review Board of National University of Singapore (NUS-IRB 10–285). Whole venous blood was collected in Citrate-Phosphate Dextrose-Adenine-1 (CPDA-1, JMS) and PBMC were isolated from whole blood over Ficoll-Paque PLUS (GE Healthcare) density gradient. Pelleted RBC were washed twice in RPMI 1640 (Sigma-Aldrich) and stored 1:1 in MCM. Remaining RBCs within the PBMC fraction were lysed with ACK lysis buffer (Life Technologies) and purified PBMC were washed twice with RPMI 1640. PBMCs were counted and cryopreserved at a concentration of $1 \times 10^8$ cells/ml in 1:1 RPMI and PBMC freezing medium of 85% fetal bovine serum (Gibco) and 15% dimethyl sulfoxide (Sigma-Aldrich) in liquid nitrogen.

**Mice**. NOD/SCID IL2γnull (NSG) mice were approved by the institutional animal care and use committee (IACUC) of National University of Singapore (NUS), Agency of Science, Technology and Research (A*STAR) and Massachusetts Institute of Technology (MIT). Mice used were female and between 6 and 10 weeks of age. Mice were housed at an ambient temperature of between 19 °C to 24 °C and humidity levels of 30% to 70% in a 12-hour light–dark cycle. Mice also had ad libitum access to food and water.

**In vivo infection**. 6–10 week-old female NSG mice were injected daily with 1 ml of RBC mixture (50% human RBC, 25% RPMI, 25% human AB serum) intraperitoneally to generate RBC-NSG mice[59]. Human RBC reconstitution was assessed every other day by determining CD235ab levels via flow cytometry. Mice with reconstitution levels above 20% were infected with $1 \times 10^7$ mixed stage *P. falciparum* intravenously. Peripheral blood parasitemia was determined via Giemsa staining. Upon detection of parasitemia, mice were bled and adapted parasites were recovered.

For macrophage depletion, RBC-NSG mice were treated with 25 μl of Clodrosome® Liposomal Clodronate (5 mg/ml) (Encapsula NanoSciences LLC) intraperitoneally on day −6, day −4, and day −2 before infection with *P.falciparum* at day 0.

In vivo imaging was performed using an IVIS® Spectrum (PerkinElmer) on mice injected with RediJect D-Luciferin Bioluminescent Substrate (PerkinElmer) at 100 μl via intraperitoneal injection. Data was analyzed on LivingImage 3.2 (PerkinElmer).

**Whole-genome sequencing**. *P. falciparum* genomic DNA was isolated using NucleoSpin® Blood Columns (Macherey-Nagel) as per manufacturer's protocol. Whole-genome next-generation sequencing of adapted and non-adapted *P. falciparum* was performed at Singapore Centre for Environmental Life Sciences Engineering Sequencing Core using Illumina Miseq Run V3. FASTQ files of sequencing reads were aligned to the *P. falciparum* 3D7 reference genome available, at PlasmoDB, using bowtie2 v2.3.2. SAM file generated from the alignment was converted to BAM files using samtools v1.3. SNP variant calling on the BAM files was done using freebayes v1.0.1 and SNP filtering based on QUAL using vcffilter.

**Transcriptional microarray analysis**. Microarray analysis was performed as described previously[34]. Briefly, *P. falciparum* RNA was isolated using TRIzol reagent (Invitrogen) as per the manufacturer's protocol. RNA integrity was determined using 2100 Bioanalyzer with RNA 6000 Nano chips (Agilent). cDNA was synthesized using a combination of SMARTer PCR cDNA Synthesis Kit (Takara) and SuperScript II Reverse Transcriptase (ThermoFisher). cDNA was then amplified in the presence of 0.225 mM amino-allyl-dUTP (Biotium) using Taq DNA polymerase (New England Biolabs). PCR products were purified with MinElute PCR purification kit (Qiagen) according to the manufacturer's protocol and eluted in 16 μl of elution buffer. The reference pool was created by adding equal amounts of RNA from each time point from the non-adapted strain.

4 μg of amplified DNA samples were labeled with Cy3 or Cy5 (GE Healthcare). Experimental samples were labeled with Cy5 while reference pools were labeled with Cy3. Cy5-labeled and Cy3-labeled reference pool were mixed and hybridized on post-processed microarray chips using the Agilent hybridization system (Agilent) and scanned using PowerScanner™ (Tecan). Scanned images were analyzed using GenePix Pro 6.0 (Axon Instruments). Microarray data was LOESS normalized and filtered for signal intensity over the background noise using R package LIMMA v3.30[60]. Differentially expressed genes were determined using Significance Analysis of Microarray (SAM)[28].

**Peripheral blood mononuclear cell (PBMC) purification**. PBMCs were isolated using Ficoll-Paque PLUS (GE Healthcare) as per manufacturer protocol. NK cells and monocytes were isolated from purified PBMCs using EasySep™ Human NK Cell Isolation Kit and EasySep™ Human Monocyte Isolation Kit (STEMCELL), respectively.

**Phagocytosis assay**. Monocyte-derived macrophages were obtained by culturing purified monocytes in RPMI 10% FBS for 7 days. Non-adherent cells are removed. Phagocytosis assay was performed by incubating DAPI-treated trophozoites at a ratio of 5 iRBCs to 1 macrophage for 90 mins. Thereafter, the co-culture was washed in PBS to remove excess iRBCs. Macrophages were detached using Accutase (StemCell Technologies), stained with anti-human CD14 (Clone 63D3; Biolegend, 1:100 dilution), and quantified using Attune NxT (Life Technologies) and analyzed using Flowjo v10. Percentage phagocytosis is calculated as the percentage of DAPI-positive macrophages.

**Primary NK cell parasitemia reduction co-culture**. Primary NK cells were co-cultured with trophozoites at a parasitemia of 0.5% and at a ratio of 1 iRBCs to 10 NK cells for 96 h. Quantification of parasitemia was done by flow cytometry by staining for NK cells with anti-human CD45 (Clone 2D1; Biolegend; 1:100 dilution) and anti-human CD56 (Clone MEM-188; Biolegend; 1:100 dilution)[6]. Parasitemia reduction was calculated as $\% \ Parasitemia \ reduction = \frac{Parasitemia_{no \ NK} - Parasitemia_{NK}}{Parasitemia_{no \ NK}} \times 100$. Neutralization of LILRB1 was performed with anti-LILRB1 antibody (500 ng/ml, R&D Systems, MAB20172).

**Immunofluorescence assay (IFA)**. Smears of late-stage parasite cultures were prepared on glass slides and methanol fixed on ice for 15 mins and air-dried. Smears were blocked in 3% bovine serum albumin (BSA; Sigma-Aldrich). Slides were incubated with primary rat anti-HA (Roche) at 1:100 and primary mouse anti-PfEMP1 ATS at 1:500 overnight. After three rounds of washing, slides were incubated with secondary goat anti-rat IgG (H + L) Alexa Fluor 488 (1:500; Invitrogen) and goat anti-mouse IgG (H + L) Alexa Fluor 647 (1:500; Invitrogen) with Hoechst 33342 (1:1000) for 1 hour at room temperature. The slides were then mounted in VECTASHIELD® Antifade mounting media (Vector Laboratories), imaged on LSM710 confocal microscope (Carl Zeiss), and analyzed on ZEN 2 (Carl Zeiss).

**Western blotting**. Late trophozoite parasites cultured with or without aTc were isolated using a 65% Percoll gradient. Recovered parasites were resuspended in Laemmli Sample Buffer (Bio-rad) and β-mercaptoethanol (Sigma-Aldrich) and loaded on a 10% Mini-PROTEAN® TGX™ Precast Protein Gel. Samples were transferred to 0.2 μm polyvinylidene difluoride (PVDF) membrane using Trans-Blot Turbo Transfer System (Bio-Rad). The membrane was blocked in 5% skim milk in 0.1% PBS-Tween (PBS-T) for 1 h at room temperature and incubated overnight with rat anti-HA tag (Roche) and mouse anti-Actin (Invitrogen) at 1:3000 in 2% Bovine Serum Albumin (BSA). The blot was then washed 3 times in PBS-T and probed with goat anti-rat HRP (Biolegend) and goat anti-mouse HRP (Biolegend) at 1:10,000 in 2% BSA PBS-T for 1 h at room temperature. The blot was imaged on ChemiDoc MP (Bio-Rad) in Clarity Max Western ECL Substrate (Bio-Rad). Western blot analysis was done using Image Lab v6.0 (Bio-Rad).

**Quantitative real-time PCR**. Total RNA was extracted from highly synchronized ring-stage parasites (18hpi) using TRIzol Reagent (Invitrogen) according to manufacturer's protocol. Complementary DNA was synthesized with SuperScript ll Reverse transcriptase (Invitrogen) using a mix of Oligo(dT)12–18 and random primers. Quantitative real-time PCR was performed with Luna® Universal qPCR Master Mix (NEB) using primer pairs for var gene subgroups[35].

**Quantification and statistical analysis**. Data are presented as the mean and standard error of the mean (SEM). Differences between paired samples were analyzed using a paired two-tailed *t* test, while unpaired samples were analyzed with Student's *t* test. Multiple comparison tests were performed using Tukey's multiple comparison test. A *p* value of < 0.05 was considered statistically significant. All calculations were performed using the GraphPad 7 software package.

**Reporting summary**. Further information on research design is available in the Nature Research Reporting Summary linked to this article.

## Data availability

Microarray raw data are deposited in the ArrayExpress database under the accession number E-MTAB-11764. DNA sequencing raw data are deposited in the NCBI Sequence Read Archive database under the accession numbers SAMN28580495, SAMN28580496, SAMN28580497, SAMN28580498, SAMN28580499, SAMN28580500 [https://www.ncbi.nlm.nih.gov/sra/PRJNA841277]. The main data that support the findings of this study are provided within the Article and Supplementary Information. Source data are provided with this paper.

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

## Acknowledgements

We thank Lan Hiong Wong and Donald Tay for their technical assistance, Farzad Olfat for administrative support and Neslihan Kaya for bioinformatics analysis as well as Professor Lars Hviid for kindly sharing with us the PAM1.4 antibody. This work was supported by the National Research Foundation of Singapore through the Singapore–MIT Alliance for Research and Technology's (SMART) Interdisciplinary Research Groups in Infectious Disease and Antimicrobial Resistance Research Program and the Singapore Ministry of Health's National Medical Research Council under its Cooperative Basic Research Grant (NMRC/CBRG/0040/2013) and Open Fund Individual Research Grant (NMRC/OFIRG/0058/2017). M.C. and W.Y. were supported by the SMART graduate fellowship while R.I.O. was supported by a SINGA graduate fellowship.

## Author contributions

M.C., W.Y., P.P., and J.C. designed the research and interpreted the data. M.C., C.F.P., and J.C.N. designed and performed the Tet-R aptamer and CRISPR editing in adapted *P. falciparum* experiments. M.C., W.Y., and R.H. performed and analyzed the microarray data. M.C., W.Y., and R.I.O. performed the macrophage phagocytosis assay, NK cell killing assay and qPCR quantification of PfEMP1 experiments. M.C., W.Y., Q.C., J.C., and P.P. designed and performed the huRBC-NSG mouse model experiments and analyzed the data. M.C. and W.Y. performed all other experiments. M.C., W.Y., J.C., and P.P. wrote the manuscript. J.C. and P.P. supervised the work.

## Competing interests

The authors declare no competing interests.
