## [Peer Review File · Nature Communications]

Reviewer comments, first round –

Reviewer #1 (Remarks to the Author):

The persistence and pathogenesis of *P. falciparum* in the human blood stream depends to a significant degree on the expression of variant surface antigens (VSAs) such as PfEMP1s and RIFINs that are positioned on the infected red cell surface. These proteins are implicated in the evasion of host immune clearance but this is very difficult to study in humans. In this manuscript, Chew et al use the NSG immunodeficient mouse model repopulated with human red blood cells, to study parasite evasion mechanisms of the innate immune system. They adapt a number of *P. falciparum* strains, which initially cannot grow in these mice and show that after adaptation the strains grow well. They then claim to show how the upregulation of the PfEMP1 called VAR2CSA, which has been shown to play an important role in pregnancy-associated malaria, is used by *P. falciparum* to escape the host phagocytic cell attack. Specifically, VAR2CSA seems to be involved in escaping macrophage phagocytosis. Surprisingly, the adapted parasites strains also evade NK cell-mediated killing, through expression of a specific RIFIN and interaction with the LILRB1 receptor. The humanized mouse approach chosen by the authors is very innovative and the data enticing. However, the specific experimental approach and conclusions drawn from the experimental data raise some major concerns listed below.

Major Concerns:

1. To study in vivo factors important for the parasite growth and survival, the authors infected the RBC-NSG mouse model with 6 strains of *P. falciparum* previously adapted in vitro and an in vitro non-adapted control of *P. falciparum*. The authors, after an initial adaptation in the RBC-NSG mouse model, further culture the parasite in vitro for 20 cycles before injecting them again into the RBC-NSG mice. The authors should explain why they decided to culture the parasites in vitro for additional 20 cycles before reinjecting them in the mice to show in vivo adaptation. Was the first adaptation not enough to ensure parasite growth in the mouse model? The in vitro culture part certainly complicates the interpretation of results. A major concern is the var2csa data, due to the in vitro culture between mouse transfers. It is known that continuous in vitro culture of *P. falciparum* induces changes in var gene expression and it might select for var2csa. Can the authors be certain the var2csa upregulation was mediated by in vivo adaptation? or is it driven by those additional 20 cycles in vitro? Do the authors have transcriptional data of the initial mouse-adapted parasites and in vitro cultures prior injection in the second round NSG mice? This is important to show.
2. One of the key results of this paper is the alleged role of the non-CD36 binding PfEMP1 VAR2CSA in escaping the elimination by host macrophages. However, the experiments are not set up to directly proof this. The authors did not create a parasite to directly test the role of VAR2CSA but rather chose a parasite with regulatable MAHRP1 protein expression. The lack of expression of MAHRP1 however blocks export of all PfEMP1s, not specifically only VAR2CSA. In addition, VAR2CSA expression was measured by anti-ATS staining, an antibody that recognizes a conserved region common to all the PfEMP1s. In the lines 302 to 304 the authors then state that the adapted parasite requires var2csa to multiply in the RBC-NSG mice but this statement it is not supported by direct evidence. The authors should directly show expression differences with VAR2CSA-specific antibodies, which have been published. In addition, they could show that a VAR2CSA knockout line (also published) cannot be selected to grow in human RBC NSG mice. Furthermore, the authors have access to the selection-linked integration approach used in the manuscript to select a PfEMP1 type C (CD36 binder). If the authors show that macrophage phagocytosis is indeed reduced by SLI selected parasites expressing either var2csa or a Type A var in opposition at their 3D7-SLI-PfEMP1 (PF3D7_0421300) it would show a direct role of PfEMP1s in macrophage modulation.
3. The second part of the paper concerns the role of a specific RIFIN in escaping NK cell-mediated killing. The authors have previously published work showing this. Thus, it is not clear what is new in this part of the paper. Strangely, the RBC NSG mouse-selected parasite strains upregulate this specific RIFIN but the NSG mice do not have any NK cells. This begs the question what selected for

expression of this specific RIFIN in the adapted strains?? Could it be another cell type? The authors should investigate this question. As the authors show, it is not likely to be macrophages.

Minor Comments:

1. Fig 1. The legend mentions panels A, B and C but the figure has only panels A and B.
2. In figure 2C please specify if it shows an average expression of genes over different time points or if a specific IDC time point was selected.
3. In figure S4 Panel B, the luciferase experiment to validate the TetR-DOZI aptamer system shows in the control without tetracycline treatment, increasing bioluminescence. Can the author comment on it?
4. In panel 4E, human monocyte derived macrophages were used for in vitro assays. The authors may also use mouse macrophages (since that is what their NSG model contains) for these experiments as there are significant differences between mouse and human macrophages including differences at the cell surface receptor level.
5. Also, a comparison between NF54CR AS v/s No aTc bars maybe included for statistical significance and discuss this in the manuscript.
6. In Table 1 the 3D7-SLI-PfEMP1 line is missing information.

Reviewer #2 (Remarks to the Author):

Chew et al examine *P. falciparum* adaptation to in vivo growth in RBC-NSG mice. They observe upregulation of VAR2CSA gene alongside down-regulation of CD36-binding-type PfEMP1 genes, and their data suggest that PfEMP1 surface expression is required for in vivo adaptation as well as in vitro evasion of macrophage phagocytosis; macrophage depletion of RBC-NSG mice with clodronate increased susceptibility to non-adapted parasites. They separately show infected erythrocytes (IE) of adapted parasites interact with LILRB1 to suppress NK cells. The study concept is interesting and the data indicate that NK cell and macrophage activity against IE is modified as parasites adapt to in vivo growth in this host.

The limitations of their model include the artificial RBC-NSG host environment, the use of parasite lines (3D7/NF54, W2, T994) that have been maintained in the laboratory for decades, and the inoculation with blood stage rather than sporozoite stage parasites. For this reason, they need to better show that their data correspond to those observed in human studies, to the degree that such data are available (eg, parasite transcriptomic studies of controlled human infections in malaria-naïve individuals, which are briefly described in the Discussion).

Specific comments

L77—"VAR2CSA ... enables parasites to escape". This is not directly shown by their assays, which do not incorporate a VAR2CSA-specific reagent. They should caveat this when describing their findings, and ideally provide better evidence for surface expression of VAR2CSA.

L142—"whole genome sequencing". Can the authors analyze and comment on any obvious genomic deletions in the 3D7/NF54 lines under study, versus the genome of the parental isolate and better yet of a wild-type non-adapted Pf genome?

L152—"3D7-B7, 3D7attB and NF54CR using microarray analysis". These lines come from the same parental line and hence do not represent a diverse panel.

L174—"Var2csa ... does not interact with CD36". VAR2CSA binds to CSA. Can the authors exclude that CSA in mouse vasculature may select CSA-binding parasites? Do the VAR2CSA-upregulated IEs bind to CSA, as would be expected if VAR2CSA is on their surface?

L205—Tables 2 and 3 show data for DE of genes. Data on actual expression values (for example

RPKM) in non-adapted and AS can be added to allow readers to compare actual expression levels. L303—"surface expression of VAR2CSA". Surface expression of VAR2CSA is never shown directly. Do VAR2CSA abs react to the IE surface? Do the parasites bind CSA?

L332—"surface expression of VAR2CSA". See above point. The antibody here apparently reacts to any PfEMP1.

L370—"human NK cells play a critical role in the control of early stages of malaria infection". The authors should reconcile this claim from their prior work with work by others (eg, Arora eLife 2018) that fails to find human NK cells lyse IE in the absence of surface-reactive antibody. Arora also used 3D7 parasites in their assays.

L389—"PF3D7_1254800 ... role in the modulation NK cell mediated killing (Omelianczyk et al., 2020)". Was that in press paper made available to the reviewers? Did the authors show that NK cell lysis of 3D7-SLI-RIFIN IEs was reduced by the anti-LILRB1? If not, such data (with appropriate controls) would be more convincing to claim that PF3D7_1254800/rifin interacts with LILRB1 to evade NK cell lysis.

L430—"VAR2CSA is expressed in non-pregnant volunteers". As above, this point needs to be expanded on, and the transcriptomes of their AS parasites analyzed for similarities/differences to those reported from human challenge studies. Given the limitations of their model, this can go a long way to support the generalizability of their findings.

L439—"surface expression of VAR2CSA". As above, not shown directly in these studies.

Reviewer #3 (Remarks to the Author):

Summary

In this study, the authors use the NSG mouse model, which is deficient in B, T and NK cells but retains macrophages, to study the adaptation of Pf-infected RBCs to first-line innate immune cells in vivo. The authors show that parasites require a period of adaptation in vivo for efficient infection. Strikingly, many of the genes that were upregulated/downregulated in the adapted parasites are members of the VSA family. In 3 parasite lines, the most upregulated var gene was var2csa. The authors show that mahrp1 knockdown, which disrupts surface PfEMP1 expression, leads to reduced parasite fitness in vivo and increased phagocytosis by macrophages in vitro. A RIFIN that is upregulated in the adapted parasites interacts with LILRB1 and reduces NK activation in vitro.

The authors address an important question on the mechanism of parasite adaptation to the immune system in vivo. While the adaptive immune response is excluded in this model, the authors use an elegant experimental design to examine how gene expression differs in adapted vs non-adapted parasites. However, more evidence is needed to support some of their claims, as detailed below.

Major Comments

The authors should specify that this study primarily looks at adaptation to macrophages in vivo, since NK cells, while tested in vitro, are absent in this model along with B and T cells. Given that the NSG mouse model does not have NK cells, it is unlikely that the LILRB1-binding RIFIN PF3D7_1254800 was upregulated to evade NK activity. Are there other cell types that express LILRB1 in this model?

The finding that VAR2CSA is upregulated in different adapted parasite lines is interesting, but perhaps unexpected given previous data on both CHMI-treated and naturally exposed humans (e.g. Milne et al. 2021 eLife, Bachmann et al. 2019 PLoS Pathog) where other PfEMP1 variants, in particular group B variants are more abundant than VAR2CSA. Could the authors suggest a reason for this difference?

More evidence is needed to support the claim that VAR2CSA is important for in vivo adaptation through macrophage evasion, at several levels:

a) First, while the authors have demonstrated that *var2csa* is the most upregulated PfEMP1 transcript in 3 different adapted parasite strains, is *var2csa* also the most abundant variant in these adapted parasites? (not in comparison to the unadapted line, but when compared to other PfEMP1 variants in the adapted line). Furthermore, at the surface level, what % of iRBCs express VAR2CSA? This should be quite straightforward to test using VAR2CSA-specific mAbs by flow cytometry. The mahrp knockdown would affect surface expression of other PfEMP1 variants, so it is difficult to judge the relative importance of VAR2CSA without these data.

b) Furthermore, the mahrp knockdown may have more effects beyond disruption of PfEMP1 export to the surface of iRBCs e.g. it also affects the morphology of Maurer's clefts and may affect the transport of other exported proteins. Does this knockdown also affect RIFIN and STEVOR export?

c) In Fig. 4E., the increased phagocytosis in mahrp parasites is unlikely to be explained by the reduction in VAR2CSA surface expression, since CD36-binding PfEMP1 variants would not have increased surface expression in place of VAR2CSA, but would likely also have reduced surface expression due to the knockdown.

In Fig. 4G, depletion of macrophages with clodronate leads to a clear increase in parasitemia, suggesting that macrophages are involved in the control of parasitemia even at early time points (the first few days). However, in a previous publication (Chen et al. 2014 PNAS), the authors show that there was no effect when a different mouse model (RICH) was depleted of monocytes/macrophages. Is there a difference between the two models that would explain this, and which do the authors think is more relevant to human infection?

Minor Comments

The authors should test binding to CD36 in the adapted vs non-adapted parasite lines to strengthen the argument on CD36 being a major factor. Can addition of recombinant CD36 inhibit phagocytosis?

If I understood correctly, the authors favor changes in transcription (i.e. var gene switching) rather than selection of a subset of parasites to account for the increased expression of *var2csa* in the adapted parasites, due to the maintenance of the adapted phenotype over >20 cycles. However, given that the rate of in vitro var switching is quite slow (~2%), it seems that either mechanism would be consistent with the data.

Point-to-point response

Reviewer #1 (Remarks to the Author):

The persistence and pathogenesis of *P. falciparum* in the human blood stream depends to a significant degree on the expression of variant surface antigens (VSAs) such as PfEMP1s and RIFINs that are positioned on the infected red cell surface. These proteins are implicated in the evasion of host immune clearance but this is very difficult to study in humans. In this manuscript, Chew et al use the NSG immunodeficient mouse model repopulated with human red blood cells, to study parasite evasion mechanisms of the innate immune system. They adapt a number of *P. falciparum* strains, which initially cannot grow in these mice and show that after adaptation the strains grow well. They then claim to show how the upregulation of the PFEMP1 called VAR2CSA, which has been shown to play an important role in pregnancy-associated malaria, is used by *P. falciparum* to escape the host phagocytic cell attack. Specifically, VAR2CSA seems to be involved in escaping macrophage phagocytosis. Surprisingly, the adapted parasites strains also evade NK cell-mediated killing, through expression of a specific RIFIN and interaction with the LILRB1 receptor. The humanized mouse approach chosen by the authors is very innovative and the data enticing. However, the specific experimental approach and conclusions drawn from the experimental data raise some major concerns listed below.

Major Concerns:

1A)

To study *in vivo* factors important for the parasite growth and survival, the authors infected the RBC-NSG mouse model with 6 strains of *P. falciparum* previously adapted *in vitro* and an *in vitro* non-adapted control of *P. falciparum*. The authors, after an initial adaptation in the RBC-NSG mouse model, further culture the parasite *in vitro* for 20 cycles before injecting them again into the RBC-NSG mice. The authors should explain why they decided to culture the parasites *in vitro* for additional 20 cycles before reinjecting them in the mice to show *in vivo* adaptation. Was the first adaptation not enough to ensure parasite growth in the mouse model? The *in vitro* culture part certainly complicates the interpretation of results.

Response

Long term *in vitro* culturing of *P. falciparum* has been shown to switch off PfEMP1 expression¹ and reduce expression of RIFINs^{2,3} compared to an *in vivo* infection. We grew the adapted parasites from RBC-NSG mice in *in vitro* culture of 20 cycles in order to show that adaptation phenotype is stable. *In vitro* expansion of the adapted parasites also enabled us to obtain enough parasites for RNA isolation for the microarray analysis and gave us confidence that the parasites used still retained the adapted phenotype. However, we have also infected mice directly with blood from infected mice without expanding the parasites in *in vitro* culture. These non-expanded adapted parasites were able to grow without delay in new RBC-NSG mice in a similar way as the *in vitro* expanded adapted parasites (Figure R1). We have incorporated this result in the Figure 1A of the revised manuscript.

Figure R1. Adapted parasites grow in new huRBC-NSG mice without delay. RBC-NSG mice were infected with non-adapted W2mef parasites. When parasitemia was evident, whole blood, including iRBCs, was taken and used directly to infect a new batch of RBC-NSG mice. Parasitemia was monitored overtime in the new RBC-NSG mice.

1B)

A major concern is the *var2csa* data, due to the *in vitro* culture between mouse transfers. It is known that continuous *in vitro* culture of *P. falciparum* induces changes in *var* gene expression and it might select for *var2csa*. Can the authors be certain the *var2csa* upregulation was mediated by *in vivo* adaptation? or is it driven by those additional 20 cycles *in vitro*? Do the authors have transcriptional data of the initial mouse-adapted parasites and *in vitro* cultures prior injection in the second round NSG mice? This is important to show.

Response

We thank the reviewer for raising this important point. As suggested, we have completed additional qRT-PCR analysis of the *var* gene expression of *in vitro* cultured non-adapted 3D7 parasites, adapted 3D7 parasites that were obtained directly from infected RBC-NSG mice, and the adapted parasites that were cultured for 2 weeks *in vitro*. Results from the qPCR analysis recapitulates the microarray analysis that non-adapted parasites primarily express *var* group B and C genes. Adapted parasites that were freshly isolated from RBC-NSG mice showed high *var2csa* expression. After culturing for 2 weeks, non-adapted parasites still showed *var* group B and C gene expression with negligible *var2csa* expression whereas adapted parasites maintain high expression of *var2csa*. These results have been included in the revised manuscript as Supplementary Figure S1G.

Figure R2. Quantitative RT-PCR analysis of non-adapted 3D7 and adapted 3D7 parasite from huRBC-NSG mice and subsequent *in vitro* culture. RNA was isolated from non-adapted parasites and huRBC-NSG mouse adapted parasites directly from infected mice. Part of the adapted parasites were also cultured in static *in vitro* culture for two weeks before RNA isolation. RNA was used for quantitative RT-PCR analysis of the *var* gene family members (Group A1-3, B1-2, C1-2, BC1-2, *var1*, *var2csa* and *var3*).

2.

One of the key results of this paper is the alleged role of the non-CD36 binding PfEMP1 VAR2CSA in escaping the elimination by host macrophages. However, the experiments are not set up to directly proof this. The authors did not create a parasite to directly test the role of VAR2CSA but rather chose a parasite with regulatable MAHRP1 protein expression. The lack of expression of MAHRP1 however blocks export of all PfEMP1s, not specifically only VAR2CSA. In addition, VAR2CSA expression was measured by anti-ATS staining, an antibody that recognizes a conserved region common to all the PfEMP1s. In the lines 302 to 304 the authors then state that the adapted parasite requires *var2csa* to multiply in the RBC-NSG mice but this statement it is not supported by direct evidence. The authors should directly show expression differences with VAR2CSA-specific antibodies, which have been published. In addition, they could show that a VAR2CSA knockout line (also published) cannot be selected to grow in human RBC NSG mice.

Response

We have used the well-characterized anti-VAR2CSA antibody, PAM1.4⁴ and can show using flow cytometry that the adapted parasites express VAR2CSA whereas non-adapted parasites do not (Figure R3). These results have been included in the revised manuscript as Figure 2E.

We agree that the suggested experiment using VAR2CSA knockout parasites would provide another level of additional support. However, we have not been able to generate such knockout ourselves or obtained it from other investigators. Therefore, we are unable to carry out the proposed experiments at this stage. Nevertheless, we feel that the body of evidence in this paper provides convincing support for a critical role of VAR2CSA in the adaptation process.

Figure R3. Staining with anti-VAR2CSA antibody, PAM1.4, shows surface expression of VAR2CSA on adapted parasites. Adapted parasites (top row) and non-adapted parasites (bottom row) were either stained with isotype control antibody or PAM1.4 antibody, followed by FITC-labeled secondary antibody and flow cytometry. Shown are staining profiles of forward scatter (FSC) versus PAM1.4. The numbers indicate percentages of parasites that are positive for PAM1.4 (VAR2CSA).

Furthermore, the authors have access to the selection-linked integration approach used in the manuscript to select a PfEMP1 type C (CD36 binder). If the authors show that macrophage phagocytosis is indeed reduced by SLI selected parasites expressing either *var2csa* or a Type A var in opposition at their 3D7-SLI-PfEMP1 (PF3D7_0421300) it would show a direct role of PfEMP1s in macrophage modulation.

Response

Indeed, our lab showed the tagging of *var* PF3D7_0421300 using SLI. However, given the conserved homology of the 3' end of *pfemp1* across its members, after numerous attempts, we are unsuccessful in utilizing SLI and the TetR-DOZI system to tag *var2csa* directly. Hence, we decided to tag *mharp1* instead to control the surface expression of the expressed VAR2CSA and PfEMP1 on the infected RBC surface.

4.

The second part of the paper concerns the role of a specific RIFIN in escaping NK cell-mediated killing. The authors have previously published work showing this. Thus, it is not clear what is new in this part of the paper. Strangely, the RBC NSG mouse-selected parasite strains upregulate this specific RIFIN but the NSG mice do not have any NK cells. This begs the question what selected for expression of this specific RIFIN in the adapted strains?? Could it be another cell type? The authors should investigate this question. As the authors show, it is not likely to be macrophages.

Response

We agree with the reviewer that we and others have previously shown that specific RIFINs are able to help the parasite escape NK cell killing through inhibitory receptor LILRB1. Considering the lack of NK cells in the NSG mouse model the upregulation of this RIFIN was surprising and we felt it was important to establish that indeed these parasites are able to escape NK cell killing. More broadly the results suggest that the upregulated RIFIN was involved in modulating other immune effector cells still present in the NSG mice. However, our data clearly indicates that the upregulated RIFIN has no impact on macrophage phagocytosis. At this stage we don't fully understand the reason for the upregulation of specific LILRB1 binding RIFINs in the NSG adapted parasites. However, it is known that monocytes also express the LILRB1 receptor and it is possible that the modulation of monocyte activation is somehow suppressed by the upregulated RIFIN. We feel that this is an interesting avenue that can be explored in the future.

We have addressed this in the discussion lines 383-388.

Minor Comments:

1.

Fig 1. The legend mentions panels A, B and C but the figure has only panels A and B.

Response

The legend has been corrected.

2.

In figure 2C please specify if it shows an average expression of genes over different time points or if a specific IDC time point was selected.

Response

This is an average expression over the six different timepoints which the microarray analysis was done across the IDC. We utilized the timecourse analysis algorithm in the R package limma.

3.

In figure S4 Panel B, the luciferase experiment to validate the TetR-DOZI aptamer system shows in the control without tetracycline treatment, increasing bioluminescence. Can the author comment on it?

Response

For this scan in the IVIS machine, we noted some background in those wells for some reason. Therefore, we concluded that the IVIS system requires a certain number of parasites to be confident that we are seeing a true signal.

4.

In panel 4E, human monocyte derived macrophages were used for in vitro assays. The authors may also use mouse macrophages (since that is what their NSG model contains) for these experiments as there are significant differences between mouse and human macrophages including differences at the cell surface receptor level.

Response

We have done phagocytosis experiments using mouse macrophages, specifically RAW 264.7 murine macrophage cell line, comparing adapted and non-adapted parasites. Phagocytosis of the adapted parasites is significantly less than the non-adapted parasites (Figure R4). These results are included in the revised manuscript as supplementary Figure S5.

Figure R4. Murine RAW 264.7 phagocytosis assay of adapted and non-adapted parasites. Non-adapted parasites and adapted 3D7attb parasites were co-cultured with murine RAW 264.7 to determine the difference in phagocytosis uptake. Data shown are mean \pm SEM from 3 independent experiments. ** $p < 0.01$.

5.

Also, a comparison between NF54CR AS v/s No aTc bars maybe included for statistical significance and discuss this in the manuscript.

Response

We have added this statistical analysis into the revised manuscript (Figure 4E). There is no significance seen between the NF54CR AS parasite and no aTc parasite. This is discussed in the revised manuscript as well.

6.

In Table 1 the 3D7-SLI-PfEMP1 line is missing information.

Response

We have added the 3D7-SLI-PfEMP1 line in Table 1.

Reviewer #2 (Remarks to the Author):

Chew et al examine *P. falciparum* adaptation to in vivo growth in RBC-NSG mice. They observe upregulation of VAR2CSA gene alongside down-regulation of CD36-binding-type PfEMP1 genes, and their data suggest that PfEMP1 surface expression is required for in vivo adaptation as well as in vitro evasion of macrophage phagocytosis; macrophage depletion of RBC-NSG mice with clodronate increased susceptibility to non-adapted parasites. They separately show infected erythrocytes (IE) of adapted parasites interact with LILRB1 to suppress NK cells. The study concept is interesting and the data indicate that NK cell and macrophage activity against IE is modified as parasites adapt to in vivo growth in this host.

The limitations of their model include the artificial RBC-NSG host environment, the use of parasite lines (3D7/NF54, W2, T994) that have been maintained in the laboratory for decades, and the inoculation with blood stage rather than sporozoite stage parasites. For this reason, they need to better show that their data correspond to those observed in human studies, to the degree that such data are available (eg, parasite transcriptomic studies of controlled human infections in malaria-naïve individuals, which are briefly described in the Discussion).

Specific comments

1.

L77—"VAR2CSA ... enables parasites to escape". This is not directly shown by their assays, which do not incorporate a VAR2CSA-specific reagent. They should caveat this when describing their findings, and ideally provide better evidence for surface expression of VAR2CSA.

Response

We have added surface expression of VAR2CSA by adapted parasites. See response to reviewer 1 point 2. We have also tuned down the statement in L77 as follows:

"Specifically, we identify that upregulation of VAR2CSA, a member of PfEMP1 that does not bind to CD36, appears to play a critical role in enabling parasites to escape from macrophage phagocytosis."

L303—"surface expression of VAR2CSA". Surface expression of VAR2CSA is never shown directly. Do VAR2CSA abs react to the IE surface? Do the parasites bind CSA?

L332—"surface expression of VAR2CSA". See above point. The antibody here apparently reacts to any PfEMP1.

L439—"surface expression of VAR2CSA". As above, not shown directly in these studies.

Response

We were able to access the well-characterized anti-VAR2CSA antibody, PAM1.4. As shown in response to reviewer 1 point 2, VAR2CSA was detected on the surface of 25% of the adapted parasites but not non-adapted parasites. These results have been included in the revised manuscript as Figure 2E.

2.

L142—"whole genome sequencing". Can the authors analyze and comment on any obvious genomic deletions in the 3D7/NF54 lines under study, versus the genome of the parental isolate and better yet of a wild-type non-adapted Pf genome?

Response

We have not done this analysis in detail but we do not note any major or obvious genetic deletions from the parasite strains that we utilized in the study compared to their parental or non-adapted/wild-type strains annotated in PlasmoDB.

L152—"3D7-B7, 3D7attB and NF54CR using microarray analysis". These lines come from the same parental line and hence do not represent a diverse panel.

Response

For this initial study, we wanted to characterize gene expression changes that are due to adaptation rather than differences due to the parasite strain, hence we chose strains that are similar to one another. It would be a good idea to expand this analysis to other strains in the future.

3.

L174—"Var2csa ... does not interact with CD36". VAR2CSA binds to CSA. Can the authors exclude that CSA in mouse vasculature may select CSA-binding parasites? Do the VAR2CSA-upregulated IEs bind to CSA, as would be expected if VAR2CSA is on their surface?

Response

With the data we have, we are unable to exclude binding of the adapted parasites to the mouse vasculature. However, based on the presence of late-stage parasites in the peripheral blood of the infected RBC-NSG mice, we do not think that the selection of VAR2CSA in the parasites is primarily due to binding to the mouse vasculature. Contrasting this to human infection, where late-stage parasites are able to bind to the vasculature, resulting in mainly the presence of early-stage parasites in the peripheral blood. This has also been noted by Angulo-Barturen *et al* that utilized a similar method of adapting *P. falciparum* parasites who also saw parasites of varying stages in the mouse peripheral blood⁵. However, Arnold *et al*, who employed the chlordonate method and did not adapt *P. falciparum*. In his model, he saw mainly early-staged parasites in the mouse peripheral blood and suggested that there could be due to sequestration of the parasite to the vasculature⁶.

4.

L205—Tables 2 and 3 show data for DE of genes. Data on actual expression values (for example RPKM) in non-adapted and AS can be added to allow readers to compare actual expression levels.

Response

The entire microarray data with the actual expression values for both the adapted and non-adapted parasite will be made available in the Supplementary Data.

5.

L370—"human NK cells play a critical role in the control of early stages of malaria infection". The authors should reconcile this claim from their prior work with work by others (eg, Arora eLife 2018) that fails to find human NK cells lyse IE in the absence of surface-reactive antibody. Arora also used 3D7 parasites in their assays.

Author's Point Response

Our lab⁷ and others⁸ have noted that donor NK cell response to infected RBCs are heterogenous. In fact, the proportion of non-responder NK cell donors is higher than that of responder, as noted by Korbel *et al* that only 11 donors out of 28 have NK cells that are responsive to iRBCs directly⁸. Differences in this response was associated with a distinct KIR allele⁹. In our previous work, we associated the heterogeneity of NK cell response to the MDA5 pathway⁷. Many labs have also shown direct NK cell response to infected RBCs without requiring antibody-dependent cellular cytotoxicity⁷⁻¹⁰ and NK cells were able to form stable contacts akin to immune synapse to the iRBCs⁸. Therefore, it could be conceivable that the donors utilized by Arora *et al* are possibly non-responders. In this manuscript, we have only selected responder NK cell for our assay to demonstrate parasitemia control of adapted and non-adapted parasites.

We have clarified the selection of responder NK cells in the revised manuscript line 292.

6.

L389—"PF3D7_1254800 ... role in the modulation NK cell mediated killing (Omelianczyk et al., 2020)". Was that in press paper made available to the reviewers? Did the authors show that NK cell lysis of 3D7-SLI-RIFIN IEs was reduced by the anti-LILRB1? If not, such data (with appropriate controls) would be more convincing to claim that PF3D7_1254800/rifin interacts with LILRB1 to evade NK cell lysis.

Response

The Omelianczyk *et al* paper has been published and is available as an open access article¹¹. We did not directly test the effects of anti-LILRB1 on 3D7-SLI-RIFIN and hope that we can expand on this study and observation in the future.

7. L430—"VAR2CSA is expressed in non-pregnant volunteers". As above, this point needs to be expanded on, and the transcriptomes of their AS parasites analyzed for similarities/differences to those reported from human challenge studies. Given the limitations of their model, this can go a long way to support the generalizability of their findings.

Response

Comparing *var2csa* expression to recent CHMI dataset, VAR2CSA has been shown to be upregulated in non-pregnant volunteers, suggesting that the role of VAR2CSA solely for binding to CSA found in the placenta may not be its only function. In Bachmann *et al.* 2019 PLoS Pathogen paper¹², the authors noted that in a patient, there was an increase in *var2csa* expression at day 23 post infection. *Var2csa* became the dominant *var* expressed, a switch from *var B* and *var B/C* expression (Results Fig 4B). This patient, L1-026, was noted to be able to control parasitemia until day 17 and is a male (Table S1). In the Milne *et al.* 2021 eLife paper¹³, *var2csa* expression was detected in all the test patients, 11 males and 8 females (Figure 4, source data 1). Therefore, Bachmann *et al.* also suggested in their discussion that the role of *var2csa* expression could extend beyond binding CSA.

We have tried to address this in lines 347-353 of the revised manuscript.

Reviewer #3 (Remarks to the Author):

Summary

In this study, the authors use the NSG mouse model, which is deficient in B, T and NK cells but retains macrophages, to study the adaptation of Pf-infected RBCs to first-line innate immune cells in vivo. The authors show that parasites require a period of adaptation in vivo for efficient infection. Strikingly, many of the genes that were upregulated/downregulated in the adapted parasites are members of the VSA family. In 3 parasite lines, the most upregulated var gene was *var2csa*. The authors show that *mahrp1* knockdown, which disrupts surface PfEMP1 expression, leads to reduced parasite fitness in vivo and increased phagocytosis by macrophages in vitro. A RIFIN that is upregulated in the adapted parasites interacts with LILRB1 and reduces NK activation in vitro.

The authors address an important question on the mechanism of parasite adaptation to the immune system in vivo. While the adaptive immune response is excluded in this model, the authors use an elegant experimental design to examine how gene expression differs in adapted vs non-adapted parasites. However, more evidence is needed to support some of their claims, as detailed below.

Major Comments

1)

The authors should specify that this study primarily looks at adaptation to macrophages in vivo, since NK cells, while tested in vitro, are absent in this model along with B and T cells. Given that the NSG mouse model does not have NK cells, it is unlikely that the LILRB1-binding RIFIN PF3D7_1254800 was upregulated to evade NK activity. Are there other cell types that express LILRB1 in this model?

Response

See response to review 1 point 4, we have specified that RBC-NSG mice lack NK cells, T cells and B cells and added discussion on possible immune cells that select for adapted parasites for VAR2CSA expression.

2)

The finding that VAR2CSA is upregulated in different adapted parasite lines is interesting, but perhaps unexpected given previous data on both CHMI-treated and naturally exposed humans (e.g. Milne et al. 2021 eLife, Bachmann et al. 2019 PLoS Pathog) where other PfEMP1 variants, in particular group B variants are more abundant than VAR2CSA. Could the authors suggest a reason for this difference?

Response

More recent CHMI dataset have shown that VAR2CSA has been upregulated in non-pregnant volunteers, suggesting that the role of VAR2CSA solely for binding to CSA found in the placenta may not be its only function. In the mentioned Bachmann et al. 2019 PLoS Pathogen paper¹², the authors noted that in a patient, there was an increase in *var2csa* expression at day 23 post infection. *Var2csa* became the dominant *var* expressed, a switch from *var B* and *var B/C* expression (Results Fig 4B). This patient, L1-

026, was noted to be able to control parasitemia until day 17 and is a male (Table S1). In the Milne et al. 2021 eLife paper¹³, *var2csa* expression was detected in all the test patients, 11 males and 8 females (Figure 4, source data 1). Therefore, Bachmann et al. also suggested in their discussion that the role of *var2csa* expression could extend beyond binding CSA.

We have tried to address this in lines 347-353 of the revised manuscript.

3)

More evidence is needed to support the claim that VAR2CSA is important for in vivo adaptation through macrophage evasion, at several levels:

3a)

First, while the authors have demonstrated that *var2csa* is the most upregulated PfEMP1 transcript in 3 different adapted parasite strains, is *var2csa* also the most abundant variant in these adapted parasites? (not in comparison to the unadapted line, but when compared to other PfEMP1 variants in the adapted line).

Author's Point response

Yes, *var2csa* is still the most abundant variant when compared among the adapted parasites based on our microarray dataset. This is also confirmed by the qRT-PCR data (see Figure R2), which has been incorporated into the revised manuscript as Figure S1G.

3b)

Furthermore, at the surface level, what % of iRBCs express VAR2CSA? This should be quite straightforward to test using VAR2CSA-specific mAbs by flow cytometry. The mahrp knockdown would affect surface expression of other PfEMP1 variants, so it is difficult to judge the relative importance of VAR2CSA without these data.

Response

We were able to determine the upregulation of VAR2CSA in the adapted parasites using the well-characterized anti-VAR2CSA antibody, PAM1.4, and show that about 25% of the adapted parasites show clear VAR2CSA expression compared to non-adapted parasites. See response to reviewer 1 point 2. These results have been included in the revised manuscript as Figure 2E.

3c)

Furthermore, the mahrp knockdown may have more effects beyond disruption of PfEMP1 export to the surface of iRBCs e.g. it also affects the morphology of Maurer's clefts and may affect the transport of other exported proteins. Does this knockdown also affect RIFIN and STEVOR export?

Response

While there is indeed a possibility that the *mahrp* knockdown affects other proteins besides PfEMP1 the study by Spycher *et al.*, 2008 has shown that MAHRP1 mainly impacts the export of PfEMP1 onto the surface of the iRBC but does not impact significantly on other proteins¹⁴. Niang *et al.*, 2014 also demonstrated that STEVOR is effectively exported onto the surface of the iRBC in these knockout parasites¹⁵. We therefore suspect that the impact of the knockdown is particularly significant for PfEMP1 trafficking to the iRBC.

We have stated this in the revised manuscript lines 228-229.

3d)

In Fig. 4E., the increased phagocytosis in *mahrp* parasites is unlikely to be explained by the reduction in VAR2CSA surface expression, since CD36-binding PfEMP1 variants would not have increased surface expression in place of VAR2CSA, but would likely also have reduced surface expression due to the knockdown.

Response

Ayi *et al* have shown that late-stage iRBC show decreased expression of CD47 compared to uninfected or ring-stage parasites¹⁶ (Figure 3). CD47 prevents RBC phagocytosis by macrophages through interaction with SIRP α as a marker of self. They have shown that by blocking SIRP α , macrophage phagocytosis of ring-stage *P. falciparum* iRBCs was significantly upregulated by not late-stage parasites. Our hypothesis is that in the absence of PfEMP1 surface expression through our inducible *mharp1* knockdown, the absence of CD47 in the late-stage parasites still leads to phagocytosis in the absence of CD36 binding PfEMP1. However, what Serghides *et al* first showed¹⁷ and again demonstrated by us here is that VAR2CSA expressing late-stage iRBCs are able to evade phagocytosis compared to iRBCs expressing CD36-binding PfEMP1.

We have addressed this in the revised manuscript lines 357.

3e)

In Fig. 4G, depletion of macrophages with clodronate leads to a clear increase in parasitemia, suggesting that macrophages are involved in the control of parasitemia even at early time points (the first few days). However, in a previous publication (Chen *et al.* 2014 PNAS), the authors show that there was no effect when a different mouse model (RICH) was depleted of monocytes/macrophages. Is there a difference between the two models that would explain this, and which do the authors think is more relevant to human infection?

Author's Point Response

There are a couple of differences comparing this paper to our previous work in the 2014 PNAS paper. The RICH mouse model reconstituted all human white blood lineage cells, including monocytes/macrophages, NK cells, T cells, B cells, dendritic cells, etc through adoptive transfer of human CD34⁺ hematopoietic stem/progenitor cells into NSG mice¹⁸. In contrast, RBC-NSG mice are constructed by repeated injection of human RBCs into NSG mice and therefore lack all human white blood cells. In our previous work, anti-human CD14 was used to deplete only human monocytes but not mouse monocytes/macrophages. In fact, anti-human CD14 clone M5E2 was chosen specifically as it is a

mouse antibody, that would be utilized mouse macrophages to clear the bound human monocytes. Our results showed that human monocytes had no effect in controlling parasitemia, in the presence of mouse monocytes/macrophages.

Minor Comments

The authors should test binding to CD36 in the adapted vs non-adapted parasite lines to strengthen the argument on CD36 being a major factor. Can addition of recombinant CD36 inhibit phagocytosis?

Response

We thank the reviewer for this suggestion. At this stage we feel that the qRT-PCR analysis convincingly demonstrated the low level of CD36 binding *var* in the adapted parasites while FACS analysis shows convincing expression of VAR2CSA on the surface of infected erythrocytes. Moreover, we provide additional data showing that in the presence of anti-CD36 antibody, phagocytosis of the non-adapted parasites is significantly reduced (Figure R5), suggesting a significant role of CD36 in mediating phagocytosis of parasites by macrophages. We have incorporated this data as Figure S6 of the revised manuscript.

Figure R5. Phagocytosis of iRBCs by human monocyte-derived macrophages in the presence of anti-CD36 antibody. Non-adapted and adapted 3D7attb parasites were used to infect RBCs. The iRBCs were co-cultured with human monocyte-derived macrophages with or without anti-CD36 blocking antibody. Phagocytosis of iRBCs was determined and shown as average \pm SEM, n=3.

If I understood correctly, the authors favor changes in transcription (i.e. var gene switching) rather than selection of a subset of parasites to account for the increased expression of var2csa in the adapted parasites, due to the maintenance of the adapted phenotype over >20 cycles. However, given that the rate of in vitro var switching is quite slow (~2%), it seems that either mechanism would be consistent with the data.

Response

We thank the reviewer for pointing this out. Indeed, both mechanisms would be consistent with the data. We have tried to make this clearer in the revised manuscript.

References

1. Zhang, Q. *et al.* From in vivo to in vitro: dynamic analysis of *Plasmodium falciparum* var gene expression patterns of patient isolates during adaptation to culture. *PLoS One* **6**, e20591 (2011).
2. Fernandez, V., Hommel, M., Chen, Q., Hagblom, P. & Wahlgren, M. Small, clonally variant antigens expressed on the surface of the *Plasmodium falciparum*-infected erythrocyte are encoded by the rif gene family and are the target of human immune responses. *The Journal of experimental medicine* **190**, 1393–1404 (1999).
3. Daily, J. P. *et al.* In vivo transcriptome of *Plasmodium falciparum* reveals overexpression of transcripts that encode surface proteins. *The Journal of infectious diseases* **191**, 1196–1203 (2005).
4. Barfod, L. *et al.* Human pregnancy-associated malaria-specific B cells target polymorphic, conformational epitopes in VAR2CSA: This article became available OnlineOpen after it was first published online on 14 December 2006 [14 February 2007]. *Molecular microbiology* **63**, 335–347 (2007).
5. Angulo-Barturen, I. *et al.* A murine model of falciparum-malaria by in vivo selection of competent strains in non-myelodepleted mice engrafted with human erythrocytes. *PloS one* **3**, e2252 (2008).
6. Arnold, L. *et al.* Further improvements of the *P. falciparum* humanized mouse model. *PloS one* **6**, e18045 (2011).
7. Ye, W. *et al.* Microvesicles from malaria-infected red blood cells activate natural killer cells via MDA5 pathway. *PLoS pathogens* **14**, e1007298 (2018).
8. Korbelt, D. S., Newman, K. C., Almeida, C. R., Davis, D. M. & Riley, E. M. Heterogeneous human NK cell responses to *Plasmodium falciparum*-infected erythrocytes. *The Journal of Immunology* **175**, 7466–7473 (2005).
9. Artavanis-Tsakonas, K. *et al.* Activation of a subset of human NK cells upon contact with *Plasmodium falciparum*-infected erythrocytes. *The Journal of Immunology* **171**, 5396–5405 (2003).
10. Mavoungou, E., Luty, A. J. F. & Kremsner, P. G. Natural killer (NK) cell-mediated cytolysis of *Plasmodium falciparum*-infected human red blood cells in vitro. *European cytokine network* **14**, 134–142 (2003).
11. Omelianczyk, R. I. *et al.* Rapid activation of distinct members of multigene families in *Plasmodium* spp. *Communications biology* **3**, 1–11 (2020).
12. Bachmann, A. *et al.* Controlled human malaria infection with *Plasmodium falciparum* demonstrates impact of naturally acquired immunity on virulence gene expression. *PLoS pathogens* **15**, e1007906 (2019).
13. Milne, K. *et al.* Mapping immune variation and var gene switching in naive hosts infected with *Plasmodium falciparum*. *Elife* **10**, e62800 (2021).

14. Spycher, C. *et al.* The Maurer's cleft protein MAHRP1 is essential for trafficking of PfEMP1 to the surface of Plasmodium falciparum-infected erythrocytes. *Molecular microbiology* **68**, 1300–1314 (2008).
15. Niang, M. *et al.* STEVOR is a Plasmodium falciparum erythrocyte binding protein that mediates merozoite invasion and rosetting. *Cell host & microbe* **16**, 81–93 (2014).
16. Ayi, K. *et al.* CD47-SIRP α interactions regulate macrophage uptake of Plasmodium falciparum-infected erythrocytes and clearance of malaria in vivo. *Infection and immunity* **84**, 2002–2011 (2016).
17. Serghides, L., Patel, S. N., Ayi, K. & Kain, K. C. Placental Chondroitin Sulfate A–Binding Malarial Isolates Evade Innate Phagocytic Clearance. *The Journal of infectious diseases* **194**, 133–139 (2006).
18. Chen, Q. *et al.* Human natural killer cells control Plasmodium falciparum infection by eliminating infected red blood cells. *Proceedings of the National Academy of Sciences* **111**, 1479–1484 (2014).

Reviewer comments, second round –

Reviewer #1 (Remarks to the Author):

The authors have satisfactorily addressed my major concerns and have revised the manuscript and figures. I look forward to publication of this study in Nature Communications.

Reviewer #2 (Remarks to the Author):

The PAM1.4 antibody data greatly improve the manuscript. While not essential, the addition of data with VAR2CSA-KO parasites would have improved the paper, and such parasites are available from multiple groups that have generated such KO parasites, so it is unfortunate the authors did not obtain these parasites to attempt infection of their humanized mice. Nevertheless, I am satisfied with the authors' responses to my comments, and expect readers will find this an interesting study.

Reviewer #3 (Remarks to the Author):

The authors have addressed most of my comments and strengthened the manuscript with additional data. My last remaining concern is that I still do not follow the link between VAR2CSA upregulation and reduced phagocytosis by macrophages. The Serghides et al. paper cited by the authors seems to link this effect to reduced binding to CD36. Here, the authors concur that the increased phagocytosis after VAR2CSA downregulation is unlikely to be due to increased expression of CD36-binding PfEMP1s and suggest that lack of CD47 may be a mechanism for phagocytosis of late-stage parasites. Unless there is a link between VAR2CSA and CD47, this suggests that VAR2CSA binds to a receptor on macrophages to inhibit phagocytosis, and that this effect is lost when VAR2CSA is absent. Do the authors have data to support this?

Point-by-point response

Reviewer #1 (Remarks to the Author):

The authors have satisfactorily addressed my major concerns and have revised the manuscript and figures. I look forward to publication of this study in Nature Communications.

Author's Response

We thank the reviewer for taking the time and effort to review our manuscript and appreciate the insightful comments during this review process.

Reviewer #2 (Remarks to the Author):

The PAM1.4 antibody data greatly improve the manuscript. While not essential, the addition of data with VAR2CSA-KO parasites would have improved the paper, and such parasites are available from multiple groups that have generated such KO parasites, so it is unfortunate the authors did not obtain these parasites to attempt infection of their humanized mice. Nevertheless, I am satisfied with the authors' responses to my comments, and expect readers will find this an interesting study.

Author's Response

We thank the reviewer for taking the time and effort to review our manuscript and hope that in the future we would be able to carry out the suggested VAR2CSA-KO parasite experiment. We appreciate the insightful comments from the reviewer during this process.

Reviewer #3 (Remarks to the Author):

The authors have addressed most of my comments and strengthened the manuscript with additional data. My last remaining concern is that I still do not follow the link between VAR2CSA upregulation and reduced phagocytosis by macrophages. The Serghides et al. paper cited by the authors seems to link this effect to reduced binding to CD36. Here, the authors concur that the increased phagocytosis after VAR2CSA downregulation is unlikely to be due to increased expression of CD36-binding PfEMP1s and suggest that lack of CD47 may be a mechanism for phagocytosis of late-stage parasites. Unless there is a link between VAR2CSA and CD47, this suggests that VAR2CSA binds to a receptor on macrophages to inhibit phagocytosis, and that this effect is lost when VAR2CSA is absent. Do the authors have data to support this?

Author's Response

We suggest that the lack of VAR2CSA surface expression leads to increased phagocytosis and one of the likely mechanisms of recognizing infected RBCs is due to the lack of CD47 compared to non-infected RBCs. However, we currently do not have data to support the link between VAR2CSA and CD47 and its effect on macrophage phagocytosis. Additionally, Sampaio *et al* has also noted that VAR2CSA expressing parasites (CS2) significantly reduced macrophage response compared to CS2 SBP1 knockout parasites,

which similarly prevents VAR2CSA from being expressed on the surface of the iRBCs¹. They suggest that surface expression of VAR2CSA has an inhibitory effect on human macrophages, concurring with what our data suggest as well. We thank the reviewer for taking the time and effort during this review process and appreciate the insightful comments.

References

1. Sampaio, N. G., Eriksson, E. M. & Schofield, L. Plasmodium falciparum PfEMP1 modulates monocyte/macrophage transcription factor activation and cytokine and chemokine responses. *Infection and immunity* **86**, e00447-00417 (2018).